# Dachshund Homolog 1: Unveiling Its Potential Role in Megakaryopoiesis and *Bacillus anthracis* Lethal Toxin-Induced Thrombocytopenia

**DOI:** 10.3390/ijms25063102

**Published:** 2024-03-07

**Authors:** Guan-Ling Lin, Hsin-Hou Chang, Wei-Ting Lin, Yu-Shan Liou, Yi-Ling Lai, Min-Hua Hsieh, Po-Kong Chen, Chi-Yuan Liao, Chi-Chih Tsai, Tso-Fu Wang, Sung-Chao Chu, Jyh-Hwa Kau, Hsin-Hsien Huang, Hui-Ling Hsu, Der-Shan Sun

**Affiliations:** 1Institute of Medical Sciences, Tzu Chi University, Hualien 97004, Taiwan; guan0223@gmail.com (G.-L.L.); hhchang@mail.tcu.edu.tw (H.-H.C.); imchenpk@gmail.com (P.-K.C.); 2Integration Center of Traditional Chinese and Modern Medicine, Hualien Tzu Chi Hospital, Buddhist Tzu Chi Medical Foundation, Hualien 97004, Taiwan; 3Department of Molecular Biology and Human Genetics, Tzu Chi University, Hualien 97004, Taiwan; isara2150@gmail.com (W.-T.L.); az0922663053@gmail.com (Y.-S.L.); annalin310@yahoo.com.tw (Y.-L.L.); michelle1007825@hotmail.com (M.-H.H.); 4Department of Obstetrics and Gynecology, Mennonite Christian Hospital, Hualien 97004, Taiwan; mchliaochiyuan@gmail.com (C.-Y.L.); estrogen@mch.org.tw (C.-C.T.); 5Department of Hematology and Oncology, Hualien Tzu Chi Hospital, Buddhist Tzu Chi Medical Foundation, Hualien 97004, Taiwan; tfwang@tzuchi.com.tw (T.-F.W.); oldguy-chu1129@umail.hinet.net (S.-C.C.); 6Department of Medicine, College of Medicine, Tzu Chi University, Hualien 97004, Taiwan; 7Buddhist Tzu Chi Stem Cells Center, Hualien Tzu Chi Hospital, Buddhist Tzu Chi Medical Foundation, Hualien 97004, Taiwan; 8Institute of Preventive Medicine, National Defense Medical Center, Taipei 11490, Taiwan; jhkau@mail.ndmctsgh.edu.tw (J.-H.K.); hhhuang37@gmail.com (H.-H.H.); hlhsu@mail.ndmctsgh.edu.tw (H.-L.H.)

**Keywords:** *Bacillus anthracis*, lethal toxin, DACH1, short hairpin RNA, megakaryocytic differentiation, megakaryopoiesis, polyploidization

## Abstract

Lethal toxin (LT) is the critical virulence factor of *Bacillus anthracis*, the causative agent of anthrax. One common symptom observed in patients with anthrax is thrombocytopenia, which has also been observed in mice injected with LT. Our previous study demonstrated that LT induces thrombocytopenia by suppressing megakaryopoiesis, but the precise molecular mechanisms behind this phenomenon remain unknown. In this study, we utilized 12-O-tetradecanoylphorbol-13-acetate (TPA)-induced megakaryocytic differentiation in human erythroleukemia (HEL) cells to identify genes involved in LT-induced megakaryocytic suppression. Through cDNA microarray analysis, we identified Dachshund homolog 1 (*DACH1*) as a gene that was upregulated upon TPA treatment but downregulated in the presence of TPA and LT, purified from the culture supernatants of *B. anthracis*. To investigate the function of DACH1 in megakaryocytic differentiation, we employed short hairpin RNA technology to knock down DACH1 expression in HEL cells and assessed its effect on differentiation. Our data revealed that the knockdown of DACH1 expression suppressed megakaryocytic differentiation, particularly in polyploidization. We demonstrated that one mechanism by which *B. anthracis* LT induces suppression of polyploidization in HEL cells is through the cleavage of MEK1/2. This cleavage results in the downregulation of the ERK signaling pathway, thereby suppressing *DACH1* gene expression and inhibiting polyploidization. Additionally, we found that known megakaryopoiesis-related genes, such as *FOSB*, *ZFP36L1*, *RUNX1*, *FLI1*, *AHR*, and *GFI1B* genes may be positively regulated by *DACH1*. Furthermore, we observed an upregulation of DACH1 during in vitro differentiation of CD34–megakaryocytes and downregulation of DACH1 in patients with thrombocytopenia. In summary, our findings shed light on one of the molecular mechanisms behind LT-induced thrombocytopenia and unveil a previously unknown role for DACH1 in megakaryopoiesis.

## 1. Introduction

*Bacillus anthracis*, the causative agent of anthrax, is a Gram-positive, nonmotile, aerobic, spore-forming, rod-shaped bacterium [1]. Although primarily affecting herbivores and cattle, anthrax gained substantial attention as a potential bioterrorism agent following the events of 9/11 in 2001 and the outbreak of injectional anthrax associated with spore-contaminated heroin in 2009–2010 [2,3]. Consequently, there has been a renewed focus on research to understand the pathogenesis of *B. anthracis*. Based on the entry route, *B. anthracis* endospores can lead to three types of human infections: cutaneous, gastrointestinal, and inhalational. All three forms of anthrax infection can progress to systemic disease, characterized by symptoms such as hypotension, thrombocytopenia, bleeding, anemia, multiorgan failure, and sudden shock [4,5,6,7]. Understanding the mechanisms underlying anthrax infection lays the groundwork for probing the intricate interactions between *B. anthracis* and host cells, a relationship further elucidated by its primary virulence factor, lethal toxin (LT), in disrupting cellular functions.

Anthrax LT, the primary virulence factor of *B. anthracis*, consists of two polypeptides: protective antigen (PA, 83 kDa) and lethal factor (LF, 90 kDa) [8,9,10]. LF is a zinc-dependent metalloprotease that cleaves the N-terminal domain of all mitogen-activated protein kinase kinases (MKKs/MEKs) from MEK1 to MEK7, except MEK5 [11]. This disruption affects three mitogen-activated protein kinase pathways: ERK (extracellular signal-regulated kinase), p38, and JNK (c-Jun N-terminal kinase) [12,13]. LF exerts its toxicity when combined with PA, which acts as a cell receptor binding component facilitating the entry of LF into cells [9,14]. Together, PA and LF form LT. Through cell culture and animal models, LT has been shown to impair the function of various cell types, including macrophages, lymphocytes, neutrophils, dendritic cells, endothelial cells, and platelets [15,16,17,18,19,20,21,22,23,24]. Our previous studies observed that platelets play a crucial role in LT-mediated mortality [20,25]. Given that thrombocytopenia is a common occurrence in patients with anthrax and LT-injected mice [7,20,25,26,27], investigating how LT affects the differentiation processes of platelet precursor megakaryocytes (MKs)-megakaryopoiesis is worthwhile.

Platelet production, or megakaryopoiesis, involves several key stages. It starts with the commitment of stem/progenitor cells in the bone marrow to the megakaryocyte lineage. These cells then undergo nuclear polyploidization and cytoplasmic maturation, leading to the release of platelets [28,29,30,31,32,33,34,35]. The process includes the differentiation of hematopoietic stem cells into megakaryoblasts and further differentiation into bipotential erythroid/megakaryocytic cells. Nuclear polyploidization, marked by specific surface markers, such as CD41 or CD61, occurs in the second stage, along with increased cell size and demarcation membrane formation [36]. Notably, polyploidization and cytoplasmic maturation follow distinct pathways [37,38,39,40]. Polyploidization involves multiple rounds of DNA replication without cytoplasmic division (endomitosis), resulting in increased ploidy levels correlated with cell size [41,42,43]. The final stage encompasses proplatelet formation and the release of functional platelets. Transcription factors, such as Runt-related transcription factor 1 (RUNX-1), GATA-1, FOG-1, NFE2, GFI1B, aryl hydrocarbon receptor (AHR), and FLI-1, are crucial to regulating megakaryopoiesis [32,33,34,35,44,45,46]. Megakaryopoiesis, the process of platelet production, involves the commitment of stem/progenitor cells to the megakaryocyte lineage, followed by nuclear polyploidization and cytoplasmic maturation, culminating in proplatelet formation and the release of functional platelets.

Previous studies indicated the involvement of the ERK pathway in the second stage of megakaryopoiesis, induced by thrombopoietin (TPO) or phorbol ester (12-O-tetradecanoylphorbol-13-acetate) (TPA) [47,48,49,50,51]. Given that LF inhibits MAPK pathways, including ERK, and considering the vital role of ERK in megakaryocytic differentiation, we hypothesized that LT may hinder megakaryocytic differentiation.

The gene Dachshund homolog 1 (*DACH1*), a human counterpart of the *Drosophila melanogaster dachshund* (*dac*) gene, encodes a chromatin-associated protein that interacts with other DNA-binding transcription factors to regulate genes crucial for determining cell fates in the eyes, legs, and nervous system of flies [52,53]. The name “dachshund” was used to describe mutant flies with extremely short legs relative to their body length, highlighting the importance of this gene in developmental processes [52,54]. DACH1 shares a high degree of similarity across various organisms. While mouse homolog Dach1 and Dach2 knock-out mice did not exhibit abnormalities in eyes, limbs, or brain, the double mutant (Dach1/2) mice showed impaired female reproductive tract development [54,55]. Furthermore, human DACH1 functions as a tumor suppressor. Low expression of DACH1 has been observed in various types of tumors, such as breast tumors, esophageal cancer, lung cancer, and endometrial cancer. This low expression is correlated with increased tumor proliferation, metastasis, tumor mutation burden, reduced prognostic survival, radio sensitivity, and chemotherapy sensitivity [56,57,58,59,60,61,62]. Furthermore, high methylation of the DACH1 promoter results in reduced expression of DACH1 in tumors [62,63]. While the functional role of DACH1 in megakaryopoiesis remains unknown, earlier studies have suggested its involvement in myeloid commitment [64]. Notably, the promoter of Dach1 exhibits hypomethylation in common myeloid progenitor cells but is silenced in common lymphoid progenitor cells [65]. 

In our study, we aimed to investigate the effect of LT on TPA-induced megakaryocytic differentiation in human erythroleukemia (HEL) cells. We employed LT and cDNA microarray analysis to uncover the underlying molecular mechanism. Our findings revealed that LT treatment hindered the differentiation process triggered by TPA. To explore the molecular mechanism further, we focused on *DACH1*, a gene that displayed upregulation upon TPA treatment but downregulation in the presence of TPA and LT. We investigated how LT blocks polyploidization through the modulation of DACH1. To gain insights into the potential role of DACH1 in megakaryopoiesis, we utilized short hairpin RNA (shRNA) knockdown technology. This method utilizes a type of RNA molecule that is frequently employed in molecular biology research to silence or reduce the expression of specific genes. To delineate its relationship with other known megakaryopoiesis-related transcription factors and its roles in MK differentiation in vitro and in vivo, we conducted quantitative reverse transcription polymerase chain reaction (qRT-PCR) analysis.

## 2. Results

### 2.1. Bacillus anthracis Lethal Toxin Could Inhibit TPA-Induced Megakaryocytic Differentiation

We found that *B. anthracis* LT suppresses megakaryopoiesis, contributing to thrombocytopenia in patients with anthrax and animal models [20,25,26,27]. To understand the underlying molecular mechanism, we used a model of TPA-induced megakaryocytic differentiation [66,67]. We purified LT based on previous studies [25,68,69,70] and assessed its cytotoxicity in J774A.1 cell, known to be sensitive to LT. A 2000-fold dilution (0.775 μg/mL, the first batch or LT) and an 800-fold dilution (0.452 μg/mL, the second batch of LT) resulted in cell viability below 50% (Appendix A) and were used for subsequent experiments. We treated human HEL cells with LT in the presence of TPA to investigate its effects on TPA-induced megakaryocytic differentiation. Untreated cells and cells treated with LT, TPA, and 0.01% dimethyl sulfoxide (DMSO) (vehicle) alone served as control groups. Our results demonstrated that neither the vehicle nor LT alone induced discernible or cytotoxic effects on HEL cells compared to the untreated control group (Figure 1 and Appendix A). However, LT treatment considerably inhibited TPA-induced megakaryocytic differentiation, evidenced by reduced CD41 (the megakaryocytic-specific surface marker) (Figure 1A,B) and CD61 (another megakaryocytic-specific surface marker) expression (Appendix A) and a decrease in the percentage of polyploid cells with a DNA content greater than 4 N (Figure 1C,D). These findings align with our previous observations in in vitro models of megakaryocytic differentiation using CD34 cells from cord blood and mononuclear cells from mouse bone marrow [25], confirming the ability of LT to suppress megakaryocytic differentiation in HEL cells.

### 2.2. DACH1 Gene Expression Was Increased during the Process of Megakaryocytic Differentiation

We conducted a cDNA microarray assay to investigate genes associated with the LT-mediated suppression of megakaryopoiesis. HEL cells were treated with TPA to induce megakaryocytic differentiation, and their gene expression was compared with untreated cells (the original data are presented in Appendix A). We also compared gene expression between cells treated with TPA alone and those treated with TPA and LT (the original data are presented in Appendix A). The analysis revealed that 365 genes were upregulated more than two-fold upon TPA treatment compared with untreated cells. However, in the presence of LT and TPA, these genes showed downregulation of more than two-fold compared wtih TPA treatment alone, suggesting their potential role in LT-mediated megakaryopoiesis suppression. From the pool of 365 genes, our primary focus was identifying transcription factors affected by the disruption of the MAPK signaling pathway caused by LT. Among these genes, we identified 14 that encode transcription factors associated with cell differentiation and development. *DACH1*, one of the 14 transcription factors identified in the cDNA microarray assay, is recognized for its involvement in eye and genital development across various species, including Drosophila, mice, and humans [52,55]. However, its specific role in megakaryopoiesis requires further investigation. To validate the cDNA microarray findings, we conducted a qRT-PCR assay to compare DACH1 mRNA expression following TPA or TPA combined with LT treatment. The results confirmed the upregulation of DACH1 mRNA expression with TPA treatment and downregulation with LT-mediated megakaryocytic differentiation suppression (Figure 2A). Western blot analysis also showed an increase in DACH1 protein levels upon TPA treatment and a decrease in DACH1 protein levels upon the combination of TPA with LT treatment (Figure 2B and Appendix A). These data strongly support the considerable upregulation of DACH1 mRNA and protein during megakaryocytic differentiation and the downregulation of both DACH1 mRNA and protein induced by LT treatment in the megakaryocytic process.

### 2.3. Knockdown of the DACH1 Gene mRNA Could Block TPA-Induced Megakaryocytic Differentiation

We conducted a shRNA knockdown assay to investigate the functional role of the *DACH1* gene in megakaryocytic differentiation. We aimed to reduce DACH1 mRNA expression levels and observe their impact on TPA-induced megakaryocytic differentiation. HEL cells were transfected with lentiviruses carrying five shRNA constructs targeting distinct regions of the *DACH1* gene (#1, #2, #3, #4, and #5 shDACH1). Our preliminary data indicated that #4 and #5 shDACH1 exhibited higher knockdown efficiency than the other three shDACH1 constructs, and we subsequently employed #4 and #5 shDACH1 in the following experiments. To prevent the discrepancy causing by the random integration of the vector containing shRNA, cells transfected with lentiviruses carrying shRNA targeting the nonmammalian gene (β-galactosidase) (shLacZ) served as the control groups. The knockdown efficiency was assessed by measuring *DACH1* gene mRNA expression levels using qRT-PCR analysis, confirming a considerable reduction compared with shLacZ-transfected groups (Figure 3A). To evaluate the effect of the *DACH1* gene knockdown on TPA-induced megakaryocytic differentiation, we analyzed CD41-specific megakaryocytic differentiation markers and DNA contents using flow cytometry. Our data revealed that the knockdown of the *DACH1* gene had no considerable effect on surface marker expression (Figure 3B,C). However, it considerably blocked TPA-induced megakaryocytic differentiation–polyploidization (Figure 3D,E) without causing any cytotoxic effect (as evidenced by the absence of a significant increase in SubG1 cells compared with the shLacZ group) (Figure 3D and Appendix A). These results demonstrated the crucial role the *DACH1* gene plays in megakaryopoiesis, particularly in polyploidization.

### 2.4. ERK Signaling Pathway Played a Role in DACH1-Mediated Suppression of Megakaryocytic Differentiation Induced by LT

We demonstrated that *DACH1* gene mRNA and protein expression levels are upregulated during megakaryocytic differentiation. Knocking down *DACH1* gene expression results in the blockade of megakaryopoiesis, particularly in polyploidization. We aimed to investigate the effect of LT on *DACH1* gene expression. In a previous study, we found that the ERK pathway plays a role in in vitro megakaryocytic differentiation using human umbilical cord blood hematopoietic stem cells. Additionally, we observed that LT can inhibit the ERK pathway, suppressing megakaryocytic differentiation [25]. We performed Western blot assay to determine if the ERK pathway is involved in TPA-induced megakaryocytic differentiation in HEL cells. Our results showed phosphorylation of ERK (p-ERK) of TPA treatment (Figure 4B,D and Appendix A). LT treatment led to the cleavage of MEK-1 and MEK-2 (Figure 4A,C, Appendix A). To confirm the involvement of the ERK pathway in LT-induced suppression of megakaryopoiesis, we used U0126, an inhibitor of MEK-1 and MEK-2. Our data showed that U0126 treatment could block TPA-induced polyploidy, similar to the effect of LT (Figure 1D). Notably, the expression of the CD41-specific surface maker was not inhibited but increased (Figure 1B). Furthermore, in the presence of U0126, the upregulated mRNA expression of the *DACH1* gene induced by TPA was downregulated (Figure 4E). These findings suggest that the inhibition of the ERK pathway and downregulation of *DACH1* gene expression are one of the mechanisms by which LT blocks megakaryopoiesis, specifically polyploidization.

### 2.5. Relationship between the DACH1 Gene and Other Megakaryopoiesis-Related Genes

Given that the *DACH1* gene represents a newly discovered transcription factor in megakaryopoiesis, our objective was to investigate its associations with other transcription factors (*FOSB* and *ZFP36L1*) identified through our microarray assay and the established transcription factors (*RUNX1*, *FLI1*, *AHR*, *GATA1*, *NFE2*, and *GFI1B*) known to participate in megakaryopoiesis. This exploration aimed to ascertain whether DACH1 plays a pivotal role as an early-determining gene in megakaryopoiesis. Our hypothesis suggests that genes positively regulated by DACH1 will exhibit downregulation upon *DACH1* gene knockdown, while genes negatively regulated by DACH1 will show upregulation following *DACH1* gene knockdown. Additionally, we anticipate that genes upstream or unrelated to the *DACH1* gene will show no change in the expression level following the *DACH1* gene knockdown. Given the high knockdown efficiency observed in Figure 3A, we proceeded with #4 shDACH1. We conducted qRT-PCR analysis to assess the relative fold change in mRNA expression levels of all genes within the shDACH1-TPA treated group. We used the mRNA expression level of each gene in the shLacZ-TPA treated group as the reference (set at one-fold). Our findings indicate that the relative expression levels of *FOSB*, *ZFP36L1*, *RUNX1*, *FLI1*, *AHR*, and *GFI1B* were downregulated upon *DACH1* gene knockdown. However, the relative mRNA expression levels of *GATA1* and *NFE2* remained unchanged following the *DACH1* gene knockdown. The results suggested that the *DACH1* gene may positively regulate the *FOSB*, *ZFP36L1*, *RUNX1*, *FLI1*, *AHR*, and *GFI1B* genes. Conversely, the *GATA1* and *NFE2* genes may be either upstream or unrelated to the *DACH1* gene (Figure 5).

### 2.6. DACH1 Gene Is Upregulated during CD34–MK In Vitro Differentiation

We established the crucial role of the *DACH1* gene in TPA-induced megakaryocytic differentiation in HEL cells. While HEL cells are equivalent to MK-erythroid progenitor cells [71], they are not representative of normal cells due to their erythroleukemia origin. To investigate further the role of the *DACH1* gene in megakaryocytic differentiation, we employed our 16-day CD34–MK in vitro differentiation model using human umbilical cord blood hematopoietic stem cells, as previously described [25]. In this model, hematopoietic stem cells differentiated into MKs over 16 days, driven by cytokines. During this differentiation process, two distinct populations (R1 and R2 regions) were identified based on cell size and granularity via flow cytometry analysis (Appendix A). The R1 region comprises viable cells, while the R2 region comprises dead cells [25]. We observed a gradual increase in the population of CD61^+^ megakaryocytes, with a concurrent rise in CD61^+^CD42b^+^ matured MKs in the R1 region (Figure 6A). Additionally, the mRNA expression levels of DACH1 increased on Day 4 and remained elevated throughout the entire in vitro differentiation (Figure 6B). These findings strongly indicate that DACH1 also plays a critical role in CD34–MK in vitro differentiation.

### 2.7. Suppressed mRNA Expression in Thrombocytopenic Patients

Given the critical role of the *DACH1* gene in megakaryopoiesis, our research was driven by curiosity to explore its involvement in clinical megakaryopoiesis defects, particularly in patients with thrombocytopenia, given its fundamental role in normal megakaryopoiesis. However, obtaining sufficient samples for analysis proved to be a formidable challenge due to the rarity of acute megakaryocytic leukemia. Nevertheless, we were able to procure a limited number of bone marrow samples from individuals with thrombocytopenia. Patients 1 and 2 (male) exhibited erythroid defects characterized by lower red blood cell (RBC) counts (normal range for men: 4.50–5.90 × 10^6^/μL; normal range for women: 4.00–5.20 × 10^6^/μL), reduced hemoglobin (Hb) levels (normal range for men: 13.5–17.5 g/dL; normal range for women: 12.0–16.0 g/dL), and decreased hematocrit (Hct) values (normal range for men: 41.0–53.0%; normal range for women: 36.0–46.0%), compared with three healthy donors (Donor 1—female, Donors 2 and 3—male). Additionally, all patients exhibited thrombocytopenia, characterized by lower platelet counts compared to healthy individuals (normal range for platelet count: 150–400 × 10^3^/μL) (Table 1). Our findings unveiled a decreasing trend in *DACH1* gene mRNA levels among patients with thrombocytopenia when compared with healthy donors (Figure 7). These results indicate a potential correlation between reduced *DACH1* gene expression and thrombocytopenia.

## 3. Discussion

This study has yielded two remarkable findings. First, we demonstrated that one mechanism by which *B. anthracis* LT induces polyploidization suppression in HEL cells is through the cleavage of MEK1/2. This cleavage results in the downregulation of the ERK signaling pathway, thereby suppressing DACH1 gene expression and inhibiting polyploidization. Second, our study presents novel evidence highlighting the critical role of DACH1 in megakaryopoiesis, particularly in polyploidization.

Thrombocytopenia, a common symptom observed in patients with anthrax, also frequently occurs in mice injected with LT [7,20,25,26,27]. However, the underlying mechanism behind this phenomenon has received limited research attention. In our previous study [25], we demonstrated that LT suppresses megakaryopoiesis by reducing the survival of differentiated MKs through the inactivation of the ERK signaling pathway and induction of apoptosis. By contrast, the current study reveals a different mechanism in which LT suppresses megakaryopoiesis by blocking differentiation. This difference in mechanisms may be attributed to the use of different models. In our previous study, we utilized in vitro MK differentiation models using mouse bone marrow cells and human umbilical cord blood cells, while the current study employed a TPA-induced MK model in HEL cells. In this study, we utilized various megakaryocytic surface markers, including CD41 for Figure 1 and Figure 3 and CD61 and CD61 combined with CD42b for Figure 6, to characterize megakaryocytic differentiation. We acknowledge that CD41 expression extends beyond megakaryocytes to include specific primitive hematopoietic cells with myeloid and lymphoid potential [72]. To ensure accurate identification of megakaryocytic differentiation from cord blood-derived CD34 cells in our study, we employed CD61 and CD42b as additional markers, as illustrated in Figure 6. However, it is essential to note that HEL cells represent a bipotential cell line, equivalent to megakaryocyte–erythroid progenitor cells [71]. When differentiating into megakaryocytes, it is common practice to utilize CD41 or CD61 for identification without concern for indistinguishability with primitive hematopoietic cells with myeloid and lymphoid potential. Although we lack CD61 data for Figure 3, we included Figure 1 with CD61 data in Appendix A for further clarification.

Although this study identified the mechanism by which LT suppresses megakaryopoiesis in HEL cells, namely, through the cleavage of MEK1/2, resulting in the inactivation of the ERK pathway, downregulation of DACH1, and inhibition of polyploidization, this represents just one of the mechanisms involved. Several other aspects warrant further investigation. First, it is essential to note that previous studies demonstrated LT ability to cleave the N-terminal domain of all mitogen-activated protein kinase kinases (MKKs/MEKs) from MEK1 to MEK7, except MEK5. Thus, it is imperative to consider the potential roles of MEK3, MEK4, MEK6, and MEK7 in LT-induced thrombocytopenia, as these MEK isoforms may also contribute to the observed effects. Second, there is evidence from several papers suggesting that LT might have other substrates beyond MEKs [14,73,74]. These studies identified sequences resembling the putative consensus cleavage site of lethal factor (LF) in various proteins found within protein databases [74]. This indicates that LT may influence megakaryopoiesis through the cleavage of substrates beyond the MEK signaling pathway. Third, how LT affects cytoplasmic maturation, specifically the expression of surface markers in MKs, must be examined. Fourth, while this study suggests that DACH1 is downregulated due to ERK signaling pathway inactivation, whether DACH1 is a direct target of the ERK signaling pathway must be determined. Last, the role of DACH1 in polyploidization and its influence on megakaryopoiesis must be explored in great detail. Addressing these questions will provide a comprehensive understanding of the complex mechanisms underlying LT-mediated suppression of megakaryopoiesis in HEL cells.

The *DACH1* gene has been extensively studied and recognized as a crucial transcription regulator for the development of various organs and tissue, including the eyes, legs, nervous system, arterial vessels, and female reproductive tract in humans, mice, and Drosophila [52,53,54,55,75,76,77]. Furthermore, DACH1 has been recognized as a tumor suppressor gene in humans [56,57,58,59,60,61,62]. Despite its extensive study, limited research directly explored its role in hematopoiesis, with only two notable studies. The first study generated Dach1-GFP reporter mice and identified a unique subpopulation within lymphoid-primed multipotent progenitors which exhibited low or negligible classic myeloid potential but had lymphoid potential. However, this study did not specifically demonstrate the importance of DACH1 in megakaryopoiesis [64]. The second study discovered that the *Dach1* gene was hypomethylated and expressed in common myeloid progenitor cells while being silenced in common lymphoid progenitor cells. This discovery led the authors proposed its involvement in myeloid commitment [65]. Nonetheless, the precise role of Dach1 in hematopoiesis, including megakaryopoiesis, remains largely uncharacterized. While neither of these studies directly supports the function of *DACH1* in megakaryopoiesis, their findings indirectly align with the results of our research.

Previous studies demonstrated distinct pathways for polyploidization and cytoplasmic maturation [37,38,39,40], thus, it is unsurprising that the knockdown of the *DACH1* gene specifically affects polyploidization. As indicated by knockout mouse studies, transcription factors Gata-1, Runx1, and Ahr are well-documented in MK polyploidization [44,45,46]. Gata-1 directly regulates cyclin D1, controlling MK polyploidization. This regulation is evident from restoring the polyploidization defect in Gata-1 knockout mice through the overexpression of cyclin D1-CDK4 [78,79]. In contrast to Gata-1, Runx1 promotes MK polyploidization by suppressing the expression of non-muscle myosin heavy chain IIB (MYH10), a critical factor for the transition from mitosis to endomitosis [80]. This transition facilitates MK polyploidization. While the precise mechanism by which Ahr controls MK polyploidization remains unclear, a potential pathway has been suggested involving the activation of Hairy and Enhancer of Split homolog-1 (HES1), a direct target gene of Ahr, which, in turn, regulates cell cycle progression [46,81,82]. Our results suggest that the *DACH1* gene may positively regulate the *FOSB*, *ZFP36L1*, *RUNX1*, *FLI1*, *AHR*, and *GFI1B* genes. However, the exact mechanism underlying this relationship requires further investigation.

In summary, our findings demonstrate that DACH1 functions as a novel transcription factor crucial for megakaryocyte (MK) polyploidization and is susceptible to targeting by *B. anthracis* LT. The observed decrease in DACH1 expression following LT treatment likely contributes to the inhibition of polyploidization while not affecting cytoplasmic maturation (surface marker expression) in the TPA-induced megakaryocytic differentiation model in HEL cells. These results provide valuable insights into the mechanisms underlying LT-mediated interference with polyploidization. However, further investigation is required to elucidate fully LT impact on cytoplasmic maturation.

## 4. Materials and Methods

### 4.1. TPA-Induced Megakaryocytic Differentiation in HEL Cells

HEL92.1.7 (ATCC No.TIB-180) cells were cultured in RPMI 1640 medium (Hyclone, Logan, UT, USA) supplemented with 10% fetal bovine serum (Biological Industries, Kibbutz Beit Haemek, Israel), 2 mM L-glutamine, 1 × non-essential amino acid, and 100 U/mL penicillin/streptomycin at 37 °C in a humidified incubator with 5% CO_2_. A total of 1 × 10^6^ HEL cells per T25 flask were cultured in the presence of 10^−8^ M TPA (Sigma, St. Louis, MO, USA), or a combination of TPA and *B. anthracis* LT, or 10 μM U0126 (#9903, Cell signaling, Danvers, MA, USA) for 3 days. The LT was purified on the basis of previous studies [25,68,69,70]. The cytotoxicity of LT was examined in the J774A.1 cell line and is shown in Appendix A (for the first batch of LT) and Appendix A (for the second batch of LT).

### 4.2. Flow Cytometry Assay

#### 4.2.1. CD Marker Characterization

To analyze the surface marker expression of megakaryocytes, the cells were initially blocked with 5% bovine serum albumin (BSA) in 1 × phosphate-buffered saline (PBS) for 1 h at 37 °C. Subsequently, the cells were stained with 5 μL of fluorescein isothiocyanate (FITC)-conjugated anti-human CD41b antibody (#348093, BD Bioscience, Franklin Lakes, NJ, USA) or 5 μL of FITC-conjugated anti-human CD61 (#555753, BD Bioscience) and 5 μL of allophycocyanin (APC)-conjugated anti-human CD42b (#303912, BioLegend, San Diego, CA, USA) antibodies in 300 μL of RPMI 1640 for 1 h at 37 °C. After staining, the cells were washed twice with 1 × PBS before flow cytometry analysis.

#### 4.2.2. DNA Content Analysis

The cells were fixed with 70% ethanol at −20 °C overnight to determine the DNA content of HEL cells. Subsequently, the cells were stained with propidium iodide (PI) staining buffer (20 μg/mL PI, 0.1% Triton X-100, 0.2 mg/mL RNaseA) for 30 min at room temperature. The cells were then washed twice with 1 × PBS before flow cytometry analysis. Flow cytometry was analyzed using a FACSCalibur instrument (Becton Dickinson, Franklin Lakes, NJ, USA).

### 4.3. RNA Isolation and cDNA Microarray Assay

Total RNA was isolated using a TRIzol reagent kit (Invitrogen, Carlsbad, CA, USA). RNase-free DNase (1 U/µg RNA; Promega, Madison, WI, USA) was applied to eliminate potential contamination from genomic DNA. RNA quality checking, including assessment of the OD260/280 ratio (above 1.8), OD260/230 ratio (above 1.5), and RNA integrity number (above 6), were passed before cDNA microarray assay. The Phalanx Biotech Group (Hsinchu, Taiwan) conducted the cDNA microarray analyses using the Human OneArray system, comprising 32,050 60-mer sense-strand oligonucleotides. These oligonucleotides correspond to 30,968 human genome probes and 1082 experimental control probes.

### 4.4. Quantitative Reverse Transcription Polymerase Chain Reaction

RNA was reverse-transcribed into cDNA using the SuperScript III First-Strand Synthesis System (Invitrogen, Hercules, CA, USA) or iScript cDNA Synthesis Kit (Bio-Rad, Hercules, CA, USA) following the manufacturers’ instructions. mRNA levels were measured through real-time PCR analysis employing SYBR Green detection (Power SYBR^®^ Green PCR, Applied Biosystems, Foster City, CA, USA) with the ABI PRISM 7300 Real-Time PCR system, LightCycler^®^ 480 Instrument II (Roche, Mannheim, Germany), or StepOnePlus™ Real-Time PCR System (Applied Biosystems™). The thermal cycling conditions used in this study were an initial denaturation step at 95 °C for 10 min, followed by 40 amplification cycles. Each cycle consisted of denaturation at 95 °C for 15 s, annealing at a temperature specific to the primers used for 30 s, and extension at 72 °C for 30 s. The annealing temperature varied depending on the primer sequences employed. The results of real-time RT-PCR were analyzed using the ABI7300 Real-time PCR System SDS Software (Version 1.2.3, Applied Biosystems). Gene copy number was determined using the relative quantitative comparative threshold cycle method (ΔΔCt) with human glyceraldehyde 3-phosphate dehydrogenase (GAPDH) as the internal control gene. The amount of target was calculated as 2^−ΔΔCt^, where ΔΔCt = [Ct (β-actin) control group − Ct (interesting gene) control group] − [Ct (β-actin) experimental group − Ct (interesting gene) experimental group] [83]. The primer sequences used for qRT-PCR are presented in Appendix A.

### 4.5. Short Hairpin RNA Knockdown Assay

The lentiviruses carrying plasmids pLKO.1-shLacZ (TRCN0000072233) and pLKO.1-shDACH1 (#4-TRCN0000118090 and #5-TRCN0000118091) with their corresponding titers were obtained from the National RNAi Core Facility at Academia Sinica in Taipei, Taiwan. The oligonucleotide sequences can be found in Appendix A. HEL cells were transfected with shDACH1 or shLacZ control viruses on the first day to initiate the virus infection. The transfection was performed at a multiplicity of infection of 8 in a 2 mL volume, supplemented with polybrene (8 μg/mL, Sigma, St. Louis, MO, USA). After three days, HEL cells were treated with 2 μg/mL puromycin (InvivoGen, San Diego, CA, USA) for four days to eliminate untransfected cells. Afterward, 10^−8^ M TPA was added on the seventh day to induce megakaryocytic differentiation in HEL cells. After 72 h of TPA treatment, the differentiation ability of HEL cells was evaluated using flow cytometry.

### 4.6. Western Blot Analysis

After washing the cells twice with 5 mL of 1 × PBS, the cells were resuspended in 1 mL of 1 × PBS and collected in a 1.5 mL microcentrifuge tube. The tube was centrifuged at 200× *g* for 5 min, and the supernatant was discarded. The pellet was resuspended in an appropriate amount of lysis buffer prepared with ddH_2_O. The lysis buffer contained 10 mM sodium orthovanadate (Na_3_VO_4_), 1 mM phenylmethylsulfonyl fluoride, 0.7 μg/mL pepstatin, and 1 μg/mL leupeptin. The samples were then incubated on ice for 30 min. The cell lysates were collected from the supernatant by centrifugation at approximately 14,000× *g* for 15 min to remove cell debris. The protein concentration of the cell extracts was measured using the BioRad protein assay. Next, the samples were mixed with an equal volume of 2× sodium dodecyl sulfate (SDS) sample denaturation buffer (containing 100 mM Tris-HCl, pH 6.8; 4% SDS; 0.2% bromophenol blue; 20% glycerol; and 10% β-mercaptoethanol), and stored at −20 °C. For electrophoresis, 30 μg of each sample was loaded onto a 10% SDS/polyacrylamide gel electrophoresis (PAGE) after being heated for 5 min at 100 °C. The separated proteins were then transferred to a nitrocellulose membrane (HybondTM-C Extra nitrocellulose paper, Amersham, UK) using a semidry transfer cell (Bio-Rad, Hercules, CA, USA) for Western blot analysis. It was incubated in 5% non-fat dry milk (Anchor, Auckland, New Zealand) in 1 × TBST (25 mM Tris-HCl, pH 7.5; 0.87% NaCl; 0.05% Tween 20) for 1 h to block the membrane. After blocking, the membrane was incubated with primary antibodies diluted in blocking buffer overnight at 4 °C. The primary antibodies used were DACH1 (1:500, ab31588 or 1:500, ab176718, Abcam), β-actin (1:8000, #A5441, Sigma or 1:5000, #610123, Novous Biologicals, Centennial, CO, USA), MEK-1 (1:1000, #610121, BD Bioscience), MEK-2 (1:1000, sc-524 or 1:1000, sc-13159, Santa Cruz), p-ERK (1:1000, #9101, Cell Signaling), and ERK (1:5000, #610123, BD Bioscience or 1:1000, #9102, Cell Signaling). Following primary antibody incubation, the membrane was washed three times for 10 min each with a blocking buffer. Afterward, the membrane probed with MEK-1, MEK-2 (sc-13159), ERK, and β-actin, and the membrane analyzed with DACH1, MEK-2 (sc-524), and p-ERK were separately incubated with horseradish peroxidase–conjugated anti-mouse secondary antibody (dilution 1:10,000, #31430, Pierce, IL, USA) and anti-rabbit secondary antibody (dilution 1:5000, #sc-2004, Santa Cruz, CA, USA) for 1 h at room temperature, followed by two washes with 1 × TBST for 10 min each and two washes with 1 × TBS (25 mM Tris-HCl, pH 7.5; 0.87% NaCl) for 10 min each [84]. The membrane was then treated with ImmobilonTM Western Chemiluminescent HRP substrate (WBKL # S0050, Millipore, Temecula, CA, USA), and the signals were obtained and quantified using the UVP Biospectrum AC system. The gel images were inverted using ImageJ software (version 1.53e), and then the band signal intensities were captured and quantified using ImageJ software. Subsequently, the gel images were cropped utilizing Adobe Photoshop CC 2019.

### 4.7. CD34–Megakaryocytes In Vitro Differentiation

Human cord blood cells were collected from full-term infants within 6 h after birth at Mennonite Christian Hospital. Following the manufacturer’s instructions, Ficoll−Paque PLUS (GE17-1440-03, GE Healthcare Life Science, Piscataway, NJ, USA) was utilized to isolate mononuclear cells (MNCs) from cord blood. The MNCs were washed with PBS three times to remove any remaining platelets. CD34^+^ cells were isolated from cord blood MNCs using a CD34 MicroBead kit (Miltenyi Biotec, Bergisch Gladbach, Germany). Before differentiation, CD34^+^ cells were expanded for 4–5 days on umbilical cord-derived mesenchymal stem cells, as per a previous study [25]. CD34^+^ cells (5 × 10^5^ cells per six-well dish) were cultured at 37 °C in a 5% CO_2_ and 100% humidity environment in a differentiation medium to initiate megakaryocytic differentiation. The differentiation medium consisted of Iscove’s modified Dulbecco’s medium supplemented with 3% fetal bovine serum, 2 mM l-glutamine, 100 U/L penicillin, 100 mg/mL streptomycin, and cytokines, including 50 ng/mL recombinant human TPO (rhTPO) (PeproTech, Cranbury, NJ, USA), 7.5 ng/mL recombinant human interleukin-6 (rhIL-6) (PeproTech), 1 ng/mL recombinant human stem cell factor (rhSCF) (PeproTech), and 13.5 ng/mL recombinant human interleukin-9 (rhIL-9) (PeproTech) [25,85]. To maintain differentiation for 16 days, the medium was freshly prepared and replaced every 4 days. The levels of specific surface markers for megakaryocytic cells were determined using FITC-conjugated anti-human CD61 and APC-conjugated anti-human CD42b antibodies. Flow cytometry (FACSCalibur instrument, Becton Dickinson, Becton Dickinson, NJ, USA) was employed to analyze the digitized results, while the CellQuest program (Becton−Dickinson) was used for quantification.

### 4.8. Statistics

The quantifiable mean and the standard error of the mean (SEM) were calculated using GraphPad Prism (5.01). Group comparisons were performed using a two-tailed Student *t*-test. A *p*-value less than 0.05 was considered statistically significant.

## Figures and Tables

**Figure 1 ijms-25-03102-f001:**
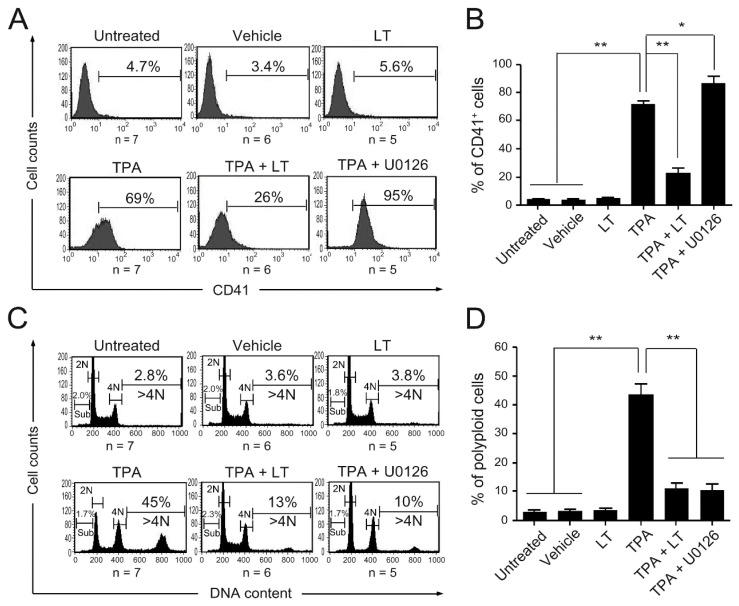
Effect of LT and U0126 on megakaryocytic differentiation. HEL cells were untreated or treated with LT, TPA (10^−8^ M), TPA combined with LT, and TPA combined with U0126 (10 μM) for three days. HEL cells treated with 0.01% DMSO served as the vehicle control. The megakaryocytic-specific surface marker (CD41) expression cell percentages were monitored (**A**) and analyzed (**B**) using flow cytometry. Cells were stained with PI, and their DNA contents were characterized (**C**) and calculated (**D**) using flow cytometry. The numbers on the top right of (**A**,**C**) represent the percentages of CD41^+^ cells and polyploidy (DNA content > 4 N) cells in each group, respectively. The numbers on the left side of (**C**) represent the percentage of SubG1 cells in each group. Data are reported as mean ± the standard error of the mean (SEM). * *p* value < 0.05, ** *p* value < 0.01, compared with the indicated groups.

**Figure 2 ijms-25-03102-f002:**
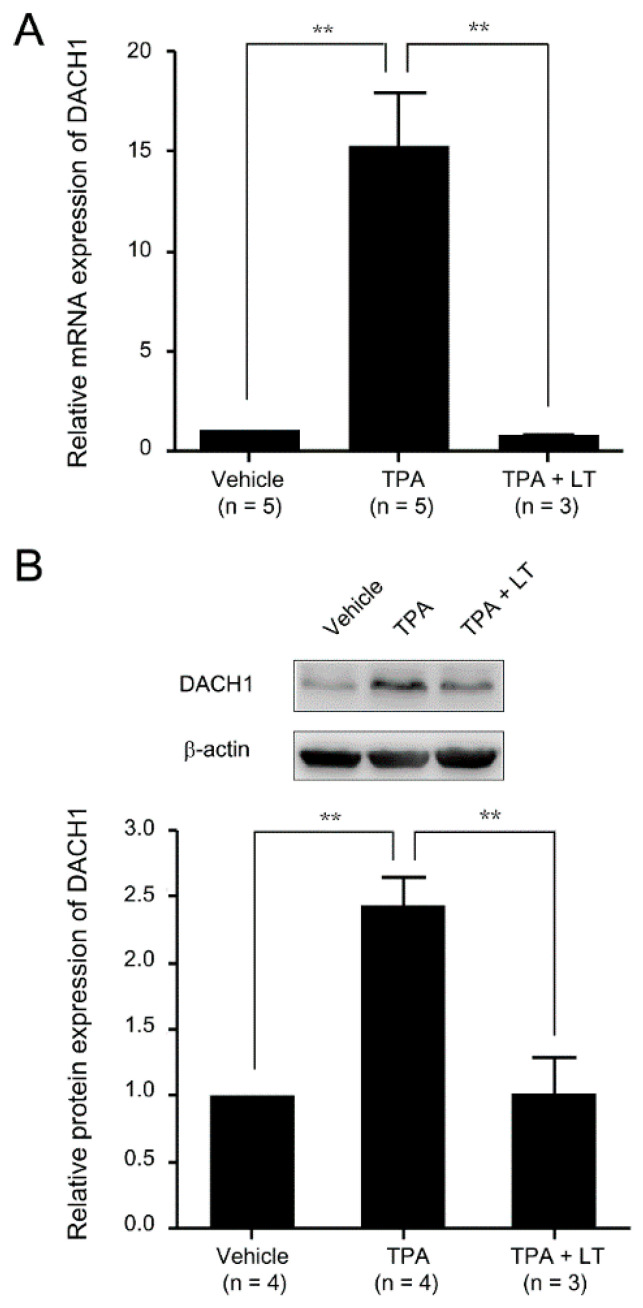
Expression level of the *DACH1* gene after megakaryocytic differentiation. (**A**) HEL cells were treated with TPA for three days or subjected to a combination of LT treatment. HEL cells treated with 0.01% DMSO served as the vehicle control. The mRNA expressions of the *DACH1* gene were analyzed using qRT-PCR and normalized to the GAPDH. The mRNA expressions of the *DACH1* gene in the vehicle group were set as one-fold. (**B**) The relative protein expressions of the *DACH1* gene were displayed using Western blot assay. β-actin served as the internal control. The protein expression level of the vehicle group was set as one-fold. The relative protein expression folds of the *DACH1* gene are shown. Full-length blots are presented in Appendix A. Data are reported as mean ± SEM. The statistical significance is denoted by ** *p* value < 0.01, compared with the indicated groups.

**Figure 3 ijms-25-03102-f003:**
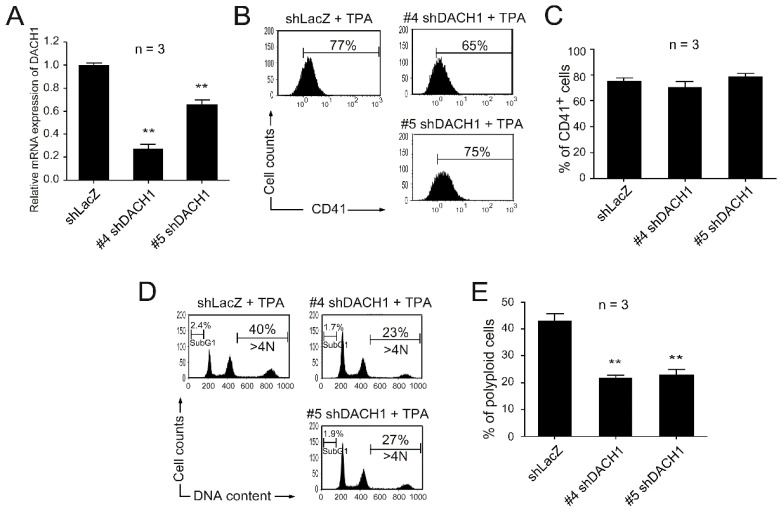
Effect on megakaryocytic differentiation after DACH1 knockdown by shRNA. HEL cells were transfected with viruses carrying shLacZ, #4 shDACH1, or #5 shDACH1, and then treated with TPA for three days to induce megakaryocytic differentiation. (**A**) The relative mRNA expression levels of the *DACH1* gene were quantified using qRT-PCR and normalized to the *GAPDH* gene. The mRNA expression levels of the *DACH1* gene in the shLacZ-transfected groups were set as one-fold. The percentages of CD41^+^ cells were characterized using flow cytometry (**B**) and analyzed (**C**). HEL cells were stained with PI, and their DNA contents were analyzed using flow cytometry (**D**) and calculated (**E**). Data are reported as mean ± SEM. ** *p* value < 0.01, compared with the shLacZ-transfected groups.

**Figure 4 ijms-25-03102-f004:**
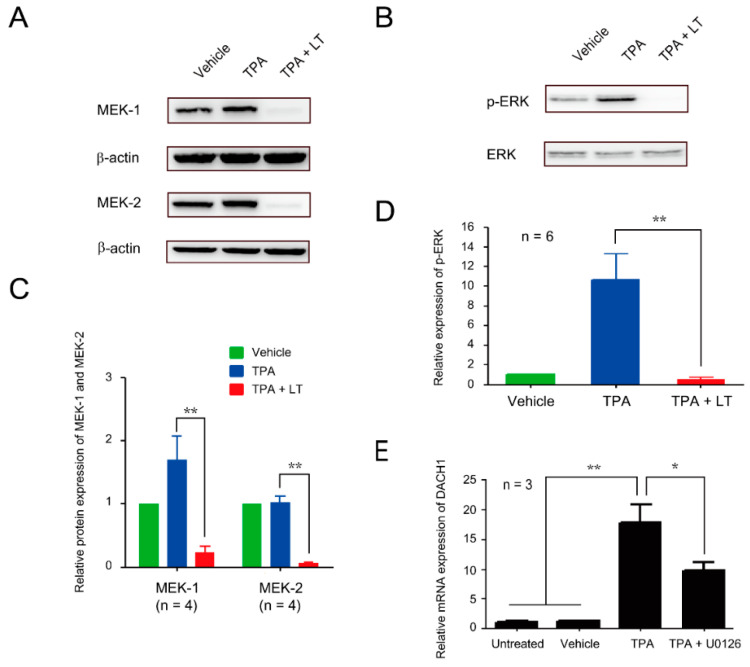
Involvement of the ERK signaling pathway in DACH1-mediated LT-induced megakaryocytic differentiation suppression. (**A**) HEL cells were treated with 0.01% DMSO (vehicle), TPA, and TPA combined with LT for 72 h. The total protein extracts were loaded on 10% SDS-PAGE. After transfer to the nitrocellulose paper, the expression levels of MEK-1 and MEK-2 were probed with antibodies against MEK-1 and MEK-2 and detected using Western blot assay, and the relative expression was analyzed (**C**). β-actin served as the internal control. Full-length blots are presented in Appendix A. (**B**) The phosphorylated-ERK (p-ERK) expression levels were detected using an antibody against p-ERK and developed through Western blot assay, and the relative expression was calculated (**D**). ERK was used as the internal control. Full-length blots are presented in Appendix A. (**E**) The relative expression levels of *DACH1* gene mRNA after TPA and U0126 treatments were determined using qRT-PCR. HEL cells untreated or treated with 0.01% DMSO (vehicle) or TPA served as control groups. The mRNA expression level of the *DACH1* genes in the untreated group was considered one-fold. Data are presented as mean ± SEM. * *p* value < 0.05, ** *p* value < 0.01, compared with the indicated groups.

**Figure 5 ijms-25-03102-f005:**
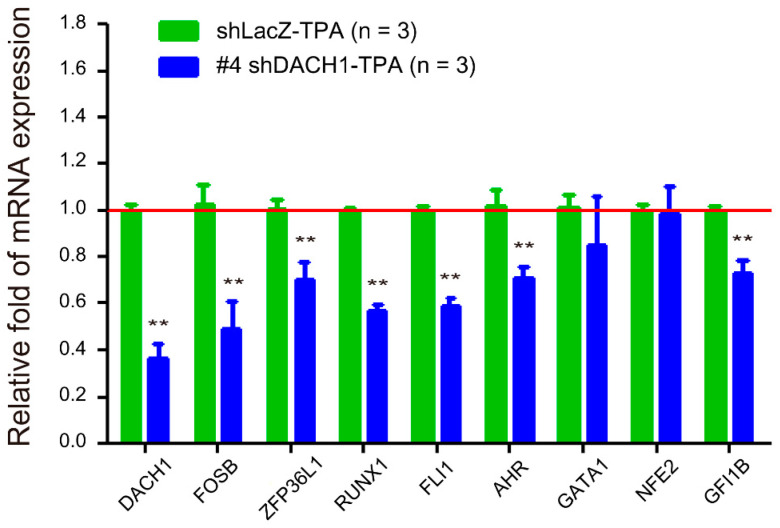
Relative mRNA expression levels of other megakaryocytic differentiation-related genes following *DACH1* gene knockdown in HEL cells. The mRNA expression levels were assessed using qRT-PCR after TPA treatment for three days and normalized to the *GAPDH* gene. The mRNA expression level of each gene in the shLacZ-TPA group was considered one-fold (red line). Data are presented as mean ± SEM. ** *p* value < 0.01 compared with the shLacZ-TPA group of each gene.

**Figure 6 ijms-25-03102-f006:**
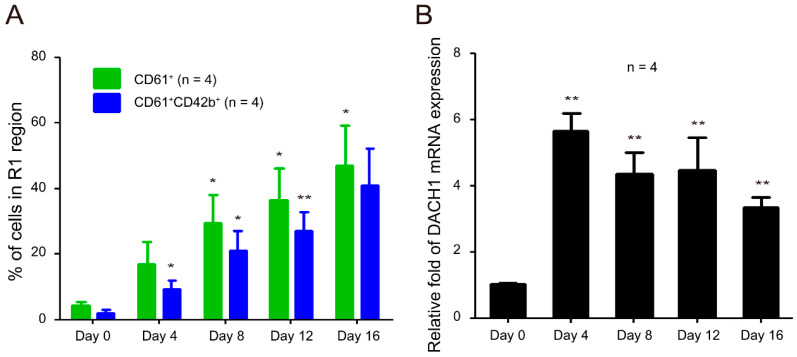
Relative mRNA expression levels of the *DACH1* gene during megakaryocytic differentiation process in the CD34–MK in vitro differentiation model. (**A**) Dynamic change in the percentages of CD61^+^ and CD61^+^CD42b^+^ cells in the R1 region. Cells were stained with antibodies against CD61 and CD42b and analyzed using flow cytometry on Days 0, 4, 8, 12, and 16. (**B**) Relative mRNA expression levels of the *DACH1* gene were analyzed using qRT-PCR and normalized to the *GAPDH* gene. The mRNA expression level of the *DACH1* genes in the Day 0 group was considered one-fold. Data are presented as mean ± SEM. * *p* value < 0.05, ** *p* value < 0.01, compared with the Day 0 group.

**Figure 7 ijms-25-03102-f007:**
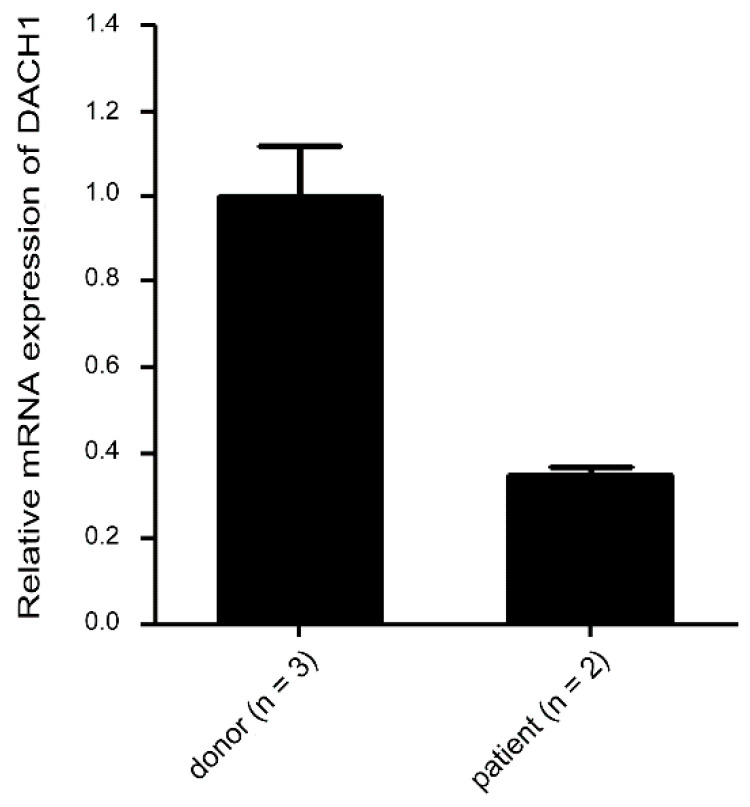
Relative mRNA expression levels of the *DACH1* gene in MNCs from healthy donors and patients with thrombocytopenia. The mRNA expressions of the *DACH1* genes were analyzed using qRT-PCR. The expression levels of the *DACH1* gene were normalized to the *GAPDH* gene. The mRNA expression levels of the *DACH1* gene in the healthy donor group were used as a reference and represented as one-fold. Data are presented as mean ± SEM.

**Table 1 ijms-25-03102-t001:** Complete blood count of healthy donors and patients.

	WBC (10^3^/μL)	RBC (10^6^/μL)	Hb (g/dL)	Hct (%)	MCV (fL)	MCH (pg)	MCHC (%)	PLT (10^3^/μL)
donor 1	6.22	4.21	13.4	39.9	94.8	31.8	33.6	420
donor 2	5.06	4.81	14.1	41.2	85.7	29.3	34.2	281
donor 3	7.01	4.70	14.1	40.5	86.2	30.0	34.8	241
patient 1	10.01	3.07 ↓	8.7 ↓	26.2 ↓	85.3	28.3	33.2	44 ↓
patient 2	4.40	3.16 ↓	10.0 ↓	30.2 ↓	95.6	31.6	33.1	143 ↓

Abbreviations: WBC = white blood cell; RBC = red blood cell; Hb = hemoglobin; Hct = hematocrit; HCV = mean corpuscular volume; MCH = mean corpuscular hemoglobin; MCHC = mean corpuscular hemoglobin concentration; PLT = platelet. ↓ indicates the levels are below the normal range.

## Data Availability

All data generated and/or analyzed during this study are available from the corresponding authors upon reasonable request.

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
