# Peer review of "Dachshund Homolog 1: Unveiling Its Potential Role in Megakaryopoiesis and Bacillus anthracis Lethal Toxin-Induced Thrombocytopenia"

_ijms, 2024, doi:10.3390/ijms25063102_

Round 1

Reviewer 1 Report (Previous Reviewer 2)

Comments and Suggestions for Authors

11. All WB data appear too clean with no background and the bands as presented do not have any reference (control tissue/mol wt marker lanes) to establish the true identity of each band. 

22. What do the authors mean by ‘The right panel displays the Western blot membrane’? Those right panels in each figure are not value added and there is no evidence that the WB band signals as shown in the left panels have been indeed acquired using the membrane as shown in the right panels.  With such noisy and dirty membranes how such a clean WB data have been generated? It looks like all WB images have been processed to eliminate the background and the non-specific signals. The authors must disclose what kind of image processing/editing has been done post acquisition of the data. 

33.  The whole purpose of showing the membrane is not clearly explained. Rather I would like to see the Ponceau S stained membrane prior primary antibody treatment to support the claim of equal loading of protein samples in each lane.

Author Response

Please see the attached file. Thank you!

Reviewer 2 Report (New Reviewer)

Comments and Suggestions for Authors

Lin et al. reported the potential role of DACH1 in LT-induced megakaryocytic suppression. Please refer to my comments: 

1) Title: Avoid abbreviations. Try to be more specific on how it links up, like "promotes" or "suppresses" the LT action.

2) Abstract: Avoid undefined abbreviations (like DACH1), remove unnecessary descriptions where you can describe further in Introduction. 

3) Introduction: Justification can be better, when is the last anthrax attack? Why are the authors particularly interested in thrombocytopenia (lines 64-66) instead of other clinically important signs and symptoms like cutaneous, GIT and respiratory anthrax? 

4) Figure 1: According to authors' speculation, LT degraded MEK protein and inhibited ERK signaling pathway that is important for megakaryocytic differentiation and polyploidy. However, based on Fig 1A, LT worked differently with MEK inhibitor (U0126) as LT inhibits megakaryocytic differentiation but not for U0126. Kindly clarify this. 

5) Fig 1 & 3: CD41 single marker is not sufficient to locate megakaryocytes, because it can be found in several hematopoietic cells as well (DOI: 10.1182/blood.V97.7.2023).  

6) Did the authors measure the cytotoxicity of LT on HEL92.1.7?

7) What are the cell number changes upon TPA treatment, with or without LT and U0126?

8) Figure 2: The authors need to attach the up- and down-regulated genes as a supplementary data (usually in Excel). 

9) Figure 2B: The quality of the band is not good (under exposed). Kindly replace with a better quality blot. 

10) Sup Fig S3: Please include at least 3 sets of western blot raw data. 

11) Fig 3: Does knockdown cause any cytotoxic effects on the cells?

12) Fig 3A & 3C Any reason where shLAZ showed significantly higher in DACH1 expression & CD41 cells? 

13) Fig 3D: Please use linear scale so that it is consistent with Fig 1C. Redo the measurement as log and linear scale will be different. 

14) Fig 4: Poor quality (under exposure) of blots. Kindly replace with other blots. 

15) Fig 4: The blots shown are not parallel with the quantitative bar. For example, MEK1 and p-ERK for TPA group (10mins). Kindly put the representative blots. 

16) Fig 4 (line 231): It is not correct to conclude the degradation started at 10mins of treatment. Why the authors measured the changes on 10mins? Any justification? 

17) Fig 4E: Is this redundant with Fig 2? 

18) Fig 5: Kindly put more info on the legend like which cells, and after how long of transfection. 

19) Fig 6A (Line 295): the use of / is not common. Kindly consider to name them as CD61+ Cd42b+. Same goes with figure and CD34-MK.  

20) Figure 6A (line 302): Kindly explain what R1 region is? Kindly include the flow dotplot or cluster plot to show the cells CD marker distribution

21) Figure 6A: Why using CD42b and 61 for this experiment but CD41 for the earlier experiment? Are the data comparable?

22) Tablw 1, fig 7 (line 309-336): Careful with this as the data are deviated away from LT-induced thrombocytopenia. I will suggest to remove this as it may partly but not help understand the effects of LT in inducing thrombocytopenia. Besides, the patient with a sample size of 2 is inconclusive in statistical analysis. 

23) Methods (line 406): Should be HEL92.1.7. 

24) Methods (Line 406): Why using HEL cells but we have better megakaryoblast cells like MEG-01 or DAMI cells? 

25) Methods (line 410): Ref 64 used 100nM and Ref 65 used 10nM. Why the authors decided to use 10nM of TPA? 

26) Methods (line 412): I am not sure why the authors tested the cytotoxicity on other cell line but not on the HEL cells. Kindly explain. 

27) Methods & Fig S1, S2: How much of LT was administrated on the HEL cells? It is not clearly explained or described. 

28) Suggest to separate the section 4.2 into "CD marker characterization" and "DNA content analysis"

29) For all the works involved in RNA or PCR, kindly include the purity and integrity data (as supplementary data). 

30) Line 427-428): How do the authors ensure there is no cell clumping in DNA content analysis? 

31) Methods (line 433-436): Were the authors outsourcing the experiment to a third party? 

32) Line 442: Any reason of using 3 different thermocycler for this experiemnt? 

33) Line 461: It should be "transfected", not "infected".

34) Line 470: Please state the G force instead of rpm. 

35) Line 474: How do the authors collect the cell lysate (extracts) from the lysis buffer (with cells)? 

36) Line 478: the blanket should include the mercaptoethanol. 

37) Line 478: Do the authors denature the samples?

38) Line 480-481: How the authors perform multiple proteins detection as nitrocellulose is not compatible with strip and reprobe?

39) Line 483: Please confirm the sources and catalogue of non-fat dry milk

40)  Line 486-488: Any reason of using multiple type of antibodies for the same protein detection? Are them comparable? 

41) Line 490-497: Kindly confirm the washing steps. 

42) Line 504: When the authors conduct this isolation? Do the authors cryopreserve the mononuclear cells? Or all the experiment are done it freshly? 

43) Line 506: What is the catalogue number for ficoll?

44) Line 508-509: What is the yield and purity of CD34+ cells? 

45) Line 522: Please confirm the flow cytometer model. 

46) Discussion: Kindly rewrite as some statements are overclaim. Line 340: it had been proven by REF 9. Carefully rewrite it. Knockdown of DACH1 did not alter the megakaryocytic differentiation (Fig 3B and 3C). Figure 3D need to prepare again by changing the X-axis scale to linear. In addition, U0126 treatment also did not alter the megakaryocytic differentiation (Fig 1). Therefore it is now questionable whether the MEK/ERK pathway is crucial or not in LT-induced thrombocytopenia. 

49) How the LT suppress DACH1 expression? 

50) Based on your current result, is the DACH1 suppressing the ERK pathway or vice versa? 

Author Response

Please see the attached file. Thank you!

Reviewer 3 Report (New Reviewer)

Comments and Suggestions for Authors

1. Abstract:

  1. The abstract is well-written, providing a clear overview of the study and its significance.
  2. Consider specifying the method used for LT purification, which could enhance the technical clarity.

2. Introduction:

  1. The introduction effectively introduces Bacillus anthracis and its lethal toxin as the focus of the study.
  2. Mentioning the rarity of acute megakaryocytic leukemia and the challenges in obtaining samples adds context to the difficulty in clinical research.
  3. The first section provides a comprehensive overview of Bacillus anthracis and its virulence factors.
  4. Consider including a transition sentence at the end of this section to smoothly lead into the discussion on lethal toxin.
  5. This section details the composition and mechanisms of anthrax LT, providing a solid foundation for the subsequent discussion.
  6. The references to previous studies effectively contextualize the toxin's impact on various cell types.
  7. Consider integrating a brief statement about the study's focus on megakaryopoiesis.
  8. The section provides a clear and concise explanation of megakaryopoiesis, establishing the basis for the study.
  9. Consider adding a sentence summarizing the primary steps in megakaryopoiesis for quick reference.
  10. This section effectively introduces the ERK pathway and its hypothesized role in LT-mediated hindrance of megakaryopoiesis.
  11. Clarify the reference to "LT" as lethal toxin for consistency.
  12. The section provides a comprehensive background on DACH1, its role in development, and its association with cancer.
  13. The connection between DACH1 and myeloid commitment is briefly mentioned, adding relevance to the current study.
  14. This section smoothly transitions to the study's objectives, linking DACH1 to megakaryocytic differentiation.
  15. Clarify the term "shRNA" upon first use for readers unfamiliar with the abbreviation.

2. Results:

1. Section 2.1 (Bacillus anthracis lethal toxin could inhibit TPA-induced megakaryocytic differentiation):

  1. The section provides a clear description of the experimental setup and results
  2. Clarify the meaning of "LT" for consistency.

2. Section 2.2 (The expression of the DACH1 gene was increased during the process of megakaryocytic differentiation):

  1. The cDNA microarray analysis is appropriately described, emphasizing the upregulation of DACH1 during megakaryocytic differentiation.
  2. The validation using qRT-PCR and Western blot provides robust confirmation of DACH1 expression changes.

3. Section 2.3 (Knockdown of the DACH1 gene mRNA could block TPA-induced megakaryocytic differentiation):

  1. The use of shRNA for DACH1 knockdown is well-documented.
  2. The flow cytometry results effectively demonstrate the impact of DACH1 knockdown on polyploidization.

4. Section 2.4 (The ERK signaling pathway played a role in DACH1-mediated suppression of megakaryocytic differentiation induced by LT):

  1. The Western blot results effectively demonstrate the involvement of the ERK pathway in LT-induced suppression.
  2. The use of U0126 as an inhibitor strengthens the link between ERK inhibition and DACH1 downregulation.

5. Section 2.5 (The relationship between the DACH1 gene and other megakaryopoiesis-related genes):

  1. The hypothesis regarding DACH1's regulatory role in other genes is logically presented.
  2. The qRT-PCR results provide supporting evidence for the relationships between DACH1 and other genes.

6. Section 2.6 (DACH1 gene is upregulated during CD34–MK in vitro differentiation):

  1. The use of a different model (CD34–MK) adds diversity to the study, reinforcing the role of DACH1 in megakaryopoiesis.

7. Section 2.7 (Suppressed mRNA expression in thrombocytopenic patients):

  1. The inclusion of clinical samples adds translational relevance to the study.
  2. Discuss the potential implications of reduced DACH1 expression in patients with thrombocytopenia.

3. Discussion:

  1. The discussion effectively summarizes the key findings and their implications.
  2. Consider addressing the limitations of the study and suggesting avenues for future research.
Comments on the Quality of English Language

Minor editing of English language required

Round 2

Reviewer 1 Report (Previous Reviewer 2)

Comments and Suggestions for Authors

I am satisfied with the revised version of the manuscript along with the responses provided by the authors.

Author Response

Thank you for the valuable suggestions provided by the reviewer. These suggestions improve the readability and clarity of our manuscript.

Reviewer 2 Report (New Reviewer)

Comments and Suggestions for Authors

Thank you the authors for the amendment. Please refer to the follow-up comments: 

Comment 4: Figure 1: According to authors' speculation, LT degraded MEK protein and inhibited ERK signaling pathway that is important for megakaryocytic differentiation and polyploidy. However, based on Fig 1A, LT worked differently with MEK inhibitor (U0126) as LT inhibits megakaryocytic differentiation but not for U0126. Kindly clarify this.

Response: We agree with reviewer’s concern. LT worked differently with MEK inhibitor (U0126). Though previous studies demonstrated that LT cleave the N-terminal domain of all mitogen-activated protein kinase kinases (MKKs/MEKs) from MEK1 to MEK7, except MEK5, several papers have proposed that LT might have other substrates [10-12]. They found that the sequences resembling the putative consensus cleavage site of lethal factor (LF) occur frequently in protein databases [12].

Follow-up comment: Kindly discuss further in the discussion. You need to update the Abstract. Please consider the potential role from other MEKs, asides of MEK 1&2. Careful with this as it may affect the interpretation of Figure 4E as well. 

Comment 5: Fig 1 & 3: CD41 single marker is not sufficient to locate megakaryocytes, because it can be found in several hematopoietic cells as well (DOI: 10.1182/blood.V97.7.2023).

Response: We appreciate the reviewer's insight regarding the limitations of using CD41 as a single marker for identifying megakaryocytes. As acknowledged, CD41 expression extends beyond megakaryocytes to include specific primitive hematopoietic cells with myeloid and lymphoid potential [13]. To address this concern, we utilized additional markers, specifically CD61 and CD42b, to identify megakaryocytic differentiation from cord blood-derived CD34 cells accurately, as depicted in Figure 6.

Furthermore, though in our experiments with HEL cells illustrated in Figure 1 and Figure 3, we employed CD41 to discern megakaryocytic differentiation, we also use CD61 to repeat the experiments and obtained consistent results. Additionally, we incorporated propidium iodide staining to highlight distinguishing characteristics such as polyploidy in megakaryocytes.

Follow-up comments: Please update the Figure 1 and 3 with CD61 data accordingly. 

Comment 6: Did the authors measure the cytotoxicity of LT on HEL92.1.7?

Response: We appreciate the reviewer's inquiry regarding the cytotoxic effects of anthrax LT on HEL92.1.7 cells. Due to the suspension nature of HEL cells, traditional cytotoxicity assays such as WST-1 or CCK-8 proliferation kits, which are commonly used for adherent cells like J774.A1 cells (as shown in our supplementary Figure S1o and Figure S2), cannot be employed. Instead, to address this concern, we conducted growth curve analyses of HEL cells under various treatments, including untreated control, DMSO vehicle control, TPA alone, LT alone, and TPA combined with LT. As illustrated in Figure B, using the trypan blue exclusion assay, our results demonstrate that LT did not exhibit cytotoxicity towards HEL cells. Notably, upon induction of differentiation by TPA (10-8 M), we observed a cessation of cell growth. However, when cells were treated with TPA in combination with LT, a partial restoration of growth was observed, albeit at a slower rate compared to LT treatment alone and the controls, as depicted in Figure A. These findings suggest that the megakaryocytic suppression induced by anthrax LT is not attributed to cytotoxicity.

Follow-up comment: WST and CCK8 assay are compatible with non-adherent cells. You may consider this in future. Current growth curve is  sufficient to demonstrate the non-cytotoxic effects of LT at that concentration. Please include this data as supplementary data. 

Comment 11: Fig 3: Does knockdown cause any cytotoxic effects on the cells?

Response: Thank you for raising this question. We conducted a propidium iodide staining assay to assess potential cytotoxic effects induced by the knockdown. Our analysis revealed no increase in the subG1 population in the knockdown groups, as determined by linear (Figure 3D) and logarithmic analyses (please see below). Therefore, we can confidently conclude that the knockdown did not induce cytotoxic effects on the cells.

Follow-up comment: Thank you the authors. Sub-G1 determination is not specific for cell death determination but it is acceptable here. Please include this as supplementary data. 

Comment 12: Fig 3A & 3C Any reason where shLAZ showed significantly higher in DACH1 expression & CD41 cells?

Response: Thank you for bringing this observation to our attention. The significantly higher expression of DACH1 mRNA and CD41 cells in the shLacZ group compared to the control was unexpected. We suspect that this discrepancy may be due to the random integration of the vector containing shRNA of LacZ into the genome of HEL cells, which served as the control for shDACH1. In contrast, the uninfected control lacks this integration issue. In response, we removed the uninfected group and utilized shLacZ as the control in Figure 3 to ensure consistency and reliability in our comparisons.

Follow-up comments: Please include this in the discussion. 

Comment 25: Methods (line 410): Ref 64 used 100nM and Ref 65 used 10nM. Why the authors decided to use 10nM of TPA?

Response: Thank you for the reviewer's suggestion. Based on prior studies and our pilot studies, we chose to induce megakaryocytic differentiation using a 10-8 M TPA treatment administered for three days to optimize results.

Follow-up comment: Please cite your prior study and attach with pilot study data if possible. 

Comment 17: Fig 4E: Is this redundant with Fig 2?

Response: We appreciate the reviewer's observation and concern regarding the potential redundancy between Figure 4E and Figure 2. While it is true that the vehicle control and TPA groups are similar between the two figures, there are important distinctions that warrant separate presentation.

In Figure 4E, we introduced the TPA plus U0126 group to investigate the potential downstream role of DACH1 in the MAPK-ERK pathway following our delineation of the pathway's involvement in LT-induced megakaryocytic suppression in HEL cells.

This additional experimental condition sheds light on the relationship between DACH1 and the MAPK-ERK pathway under specific treatment conditions. Furthermore, it is essential to note that the experiments depicted in Figure 2 and Figure 4E were conducted at different times and with distinct experimental setups. Therefore, combining and calculating the data in a single figure would not be appropriate, as each figure represents an independent experimental investigation.

In summary, while there may be similarities in experimental conditions between Figure 2 and Figure 4E, they serve distinct purposes within the context of our study. Figure 4E provides valuable insights into the downstream signaling pathways involved in LTinduced megakaryocytic suppression, specifically focusing on the role of DACH1 in the MAPK-ERK pathway. We believe that presenting these findings separately enhances the clarity and depth of our analysis. Thank you for bringing this concern to our attention, and we hope this clarification addresses any potential confusion.

Follow-up comment: Careful when interpreting this. Fig 4 (as previous studies) demonstrated LT degraded MEK1 and 2, despite with the knockdown of MERK/ERK signaling, as Fig 2 shows LT inhibited megakaryocytic differentiation but not U0126. Again, the involvement of ERK pathway in megakaryocytic differentiation is questionable. So, it may inappropriate to conclude the role of DACH1 by combining the data from U0126 and LT (Fig 4E). 

Comment 22: Tablw 1, fig 7 (line 309-336): Careful with this as the data are deviated

away from LT-induced thrombocytopenia. I will suggest to remove this as it may

partly but not help understand the effects of LT in inducing thrombocytopenia.

Besides, the patient with a sample size of 2 is inconclusive in statistical analysis.

Response: We appreciate the reviewer's feedback and concerns regarding including Table 1, Figure 7, and related text in our manuscript. In response to your suggestion, we have revised the title of our manuscript from "DACH1: A missing link in understanding megakaryocytic suppression by Bacillus anthracis lethal toxin" to "Dachshund homolog 1: Unveiling its potential role in megakaryopoiesis and Bacillus anthracis lethal toxin-induced thrombocytopenia." This change reflects the dual focus of our study: elucidating the molecular mechanisms underlying LT-induced thrombocytopenia and uncovering a previously unknown role for DACH1 in megakaryopoiesis.

Regarding the inclusion of Table 1 and Figure 7, we acknowledge the deviation from LT-induced thrombocytopenia in these analyses. However, we believe these data are still valuable as they provide clinical evidence supporting the role of DACH1 in megakaryopoiesis. Even with a small sample size, the findings from patients offer essential insights into the potential significance of DACH1 in human megakaryopoiesis.

Therefore, we have retained Table 1, Figure 7, and the related text in the revised manuscript.

We appreciate the reviewer's thoughtful consideration and understand the importance of maintaining focus in scientific analyses. We believe that the revised title and contextualization of the clinical data within the broader scope of our study adequately address these concerns while still providing valuable insights into DACH1's role in thrombocytopenia and megakaryopoiesis. Thank you for your feedback, which has helped strengthen our manuscript.

Follow-up comment: Please remove the statistical analysis as it only have 2 sample size. 

Comment 29: For all the works involved in RNA or PCR, kindly include the purity and integrity data (as supplementary data).

Response: We appreciate the reviewer's suggestion. Regrettably, providing the RNA purity and integrity data for all experiments involving RNA or PCR is not feasible. However, we can offer a representative example of how we assessed RNA purity and integrity. For instance, we evaluated RNA purity using the OD260/280 ratio (above 2.0), and we assessed RNA integrity via gel electrophoresis, which displayed intact 28S, 18S, and 5S rRNA bands (please see below).

Follow-up comment: The authors have to disclose ALL the RNA purity and integrity data especially for the RT-PCR and microarray work. Please disclose purity checking for both and RIN for microarray, at least. RIN checking is a routine step prior to microarray and I do not see any problem for not disclosing it. 

Comment 30: Line 427-428): How do the authors ensure there is no cell clumping in DNA content analysis?

Response: We appreciate the reviewer's inquiry. Cell clumping is a common occurrence in DNA content analysis. To mitigate the impact of cell clumping, we employed several measures. Firstly, before flow cytometry analysis, cells were passed through a cell mesh with a pore size of 55 μm to reduce clumping. We meticulously scrutinized the data during the analysis process to ensure that only individual, unaggregated cells were included, even with some degree of cell aggregation.

Follow-up comment: Thank you the authors. How about the clumping or duplex checking via flow cytometer? You should able to detect them and only gate the singlet for analysis. 

Comment 35: Line 474: How do the authors collect the cell lysate (extracts) from the lysis buffer (with cells)?

Response: We appreciate the reviewer's query. To address this concern, we have included the following sentence in the Methods section: "The cell extracts were collected by centrifugation at approximately 14,000 × g for 15 minutes to remove cell debris.". This addition provides clarity regarding the collection process of the cell lysate from the lysis buffer. (lines 486-487).

Follow-up comment: The authors should mention collecting the supernatants as lysate. 

Comment 39: Line 483: Please confirm the sources and catalogue of non-fat dry milk

Response: We appreciate the reviewer's inquiry. The non-fat dry milk (Anchor, NZ) used in our experiments was purchased from a local supermarket. We have modified the material in our revised manuscript (line 495).

Follow-up comment: Non-fat milk blocking is not compatible with phosphorylated protein detection (DOI: 10.4103/1947-2714.100998, 10.1111/sms.12702) because milk contains casein and will chelate away the primary antibodies. Please repeat the experiment with bovine serum albumin as blocking. 

Comment 41: Line 490-497: Kindly confirm the washing steps.

Response: We appreciate the reviewer's attention to detail. Upon reviewing our methods, we can confirm that the washing steps outlined in the manuscript are accurate and have been validated through rigorous experimental procedures. Precisely, we followed a standardized protocol for washing, including multiple washes with appropriate buffer solutions to ensure efficient removal of unbound or nonspecifically bound substances.

Follow-up comment: Please support your statement with reference or citation. 

Round 3

Reviewer 2 Report (New Reviewer)

Comments and Suggestions for Authors

Thank you the amendment. Kindly refer to the minor suggestions: 

Follow-up comment No.5: Please update the Figure 1 and 3 with CD61 data accordingly.

Response to follow-up comment 5: Thank you for the reviewer's suggestions.

We acknowledge that CD41 expression extends beyond megakaryocytes to include specific primitive hematopoietic cells with myeloid and lymphoid potential [4]. This is why, in our study, we used CD61 and CD42b as additional markers to accurately identify megakaryocytic differentiation from cord blood-derived CD34 cells, as depicted in Figure 6. However, it's important to note that HEL cells represent a bipotential cell line, equivalent to megakaryocyte-erythroid progenitor cells [5]. Upon differentiation into megakaryocytes, it is common practice to use CD41 or CD61 to identify them without concern for indistinguishability with primitive hematopoietic cells with myeloid and lymphoid potential. While we don't have CD61 data for Figure 3, we have included Figure 1 with CD61 data in the Supplementary Figure S4 for further clarification. Accordingly, the sentences "However, LT treatment considerably inhibited TPA-induced megakaryocytic differentiation, evidenced by reduced CD41 (the megakaryocytic-specific surface marker) expression (Figs. 1A and 1B) and a decrease in the percentage of polyploid cells with a DNA content greater than 4 N (Figs. 1C and 1D)." are replaced with "However, LT treatment considerably inhibited TPA-induced megakaryocytic differentiation, evidenced by reduced CD41 (the megakaryocytic-specific surface marker) (Figs. 1A and 1B) and CD61 (another megakaryocytic-specific surface marker) expression (Supplementary Figures S4) and a decrease in the percentage of polyploid cells with a DNA content greater than 4 N (Figs. 1C and 1D)." (lines 142-146).

Additional comment: Please mention this as one of the limitations of study. 

Follow-up comment No.30: Thank you the authors. How about the clumping or duplex checking via flow cytometer? You should able to detect them and only gate the singlet for analysis.

Response to follow-up comment 30: We appreciate the reviewer's continued engagement and thoughtful suggestion regarding cell clumping or duplex detection via flow cytometry. In response to this concern, we carefully assessed the flow cytometry data and applied appropriate gating strategies to ensure the analysis was focused on singlet cells. Specifically, we gated the R1 region in Figure B, as depicted by the FL2-area (A) plot and FL2-width (W), to isolate single cells for analysis. The red dots in Figure A represent these gated single cells. Conversely, the black dots in Figure A indicate cell clumps that fall outside the R1 region in Figure B. The R1 region gated from Figure B was then utilized to analyze DNA content after TPA treatment in HEL cells. This gating strategy helped us to accurately discern individual cells from clumps or duplex formations, thereby enhancing the reliability of our DNA content analysis. We thank the reviewer for their valuable input, which has strengthened the rigor and clarity of our methodology.

Additional comment: FL2-area (A) and FL2-width (W) plot is not suitable in identifying single cells. Please use FL2-A vs FL2-heigh (H) or FL2-(H) vs FL2-W. This is important as clumping cells will demonstrate higher in DNA content. Please double check the result after re-gating. 

Thank you. 

Round 4

Reviewer 2 Report (New Reviewer)

Comments and Suggestions for Authors

Thank you the authors for the amendment. Please refer to the follow-up comment: 

Follow-up comment No.30: Thank you the authors. How about the clumping or duplex checking via flow cytometer? You should able to detect them and only gate the singlet for analysis. Response to follow-up comment 30: We appreciate the reviewer's continued engagement and thoughtful suggestion regarding cell clumping or duplex detection via flow cytometry. In response to this concern, we carefully assessed the flow cytometry data and applied appropriate gating strategies to ensure the analysis was focused on singlet cells. Specifically, we gated the R1 region in Figure B, as depicted by the FL2-area (A) plot and FL2-width (W), to isolate single cells for analysis. The red dots in Figure A represent these gated single cells. Conversely, the black dots in Figure A indicate cell clumps that fall outside the R1 region in Figure B. The R1 region gated from Figure B was then utilized to analyze DNA content after TPA treatment in HEL cells. This gating strategy helped us to accurately discern individual cells from clumps or duplex formations, thereby enhancing the reliability of our DNA content analysis. We thank the reviewer for their valuable input, which has strengthened the rigor and clarity of our methodology. Additional comment: FL2-area (A) and FL2-width (W) plot is not suitable in identifying single cells. Please use FL2-A vs FL2-heigh (H) or FL2-(H) vs FL2-W. This is important as clumping cells will demonstrate higher in DNA content. Please double check the result after re-gating. Response additional comment 30:  Thank you for the reviewer’s suggestions. Based on the reviewer’s suggestion, we reanalyzed our raw data of TPA-treated HEL cells using FlowJo software (10.8.1). We identified single cells using FL2-area (A) vs. FL2-width (W) (Figure A), FL2-height (H) vs. FL2-W (Figure B), and FL2-H vs. FL2-A (Figure C), then gated the single cells and performed the DNA content analysis presented in Figures D, E, and F, respectively. The percentage of polyploid cells in each analysis was shown on the top right side of Figures D, E, and F. The data indicates no significant difference in the percentage of polyploid cells using these three different analyses.    Then we reanalyzed the percentage of polyploidy cells of TPA-treated in Figure 1C using FlowJo software with FL2-A vs. FL2- W and FL2-H vs. FL2-W plot. Our data showed that the percentages of polyploid cells using these two assay modules have no significant difference (please see the Figure below).   

Follow-up comment: 

Thank you the authors for the amendment. Sorry to say that I overlooked your earlier setting of using FL2 for gating. Please repeat the first level gating by referring to the side scatter channel (SSC-A) vs forward scatter channel (FSC-A). This indicates the cell population gating and removing debris from the analysis. The second level of gating is based on FSC-H vs FSC-A or FSC-W. This will be crucial to gate only the single cells by considering the cluster with a linear relationship. Clumped cells will demonstrate a similar FSC-H value but with an increase of FSC-A or FSC-W. Kindly note that FL1 or FL2 channels are unsuitable for gating as they do not represent the cell characteristics.

Thank you 

Author Response

Please see the attached file. Thank you!

This manuscript is a resubmission of an earlier submission. The following is a list of the peer review reports and author responses from that submission.

Round 1

Reviewer 1 Report

Comments and Suggestions for Authors

1) Why is the study focused mostly on TFs, as other proteins like cell adhesion proteins, cytoskeletal proteins, signalling ligands, etc., also play crucial roles in cell differentiation? 

2) How did the authors determine that DACH1 is the critical gene among those 14 TFs that they found from the study? Please add relevant methodology.

3) Within lines 395-396, overnight fixation of HEL cells at -20 °C in 70% ethanol may induce cell aggregation or affect expression. Do the authors have any documented protocol that they have followed? 

4) Within lines 433-434, it is stated about a MOI of 8 for virus spread. The MOI that is picked can affect how well transduction works and how much knockdown is done. It might be helpful to include details about how this MOI was found and whether pilot studies were conducted to improve transduction efficiency.

And, within lines 435-436, adding 10-8 M TPA on the seventh day is mentioned to induce megakaryocytic differentiation. It would be helpful to provide a brief rationale for the choice of TPA concentration and duration of treatment.

5) With respect to Figure 5 and results section 2.5, the authors mentioned in lines 366-367 that “DACH1 gene may positively regulate the FOSB, ZFP36L1, RUNX1, FLI1, AHR, and GFI1B genes.”

I) Why only those genes (with respect to Fig 5)?

II) Explain more about how your results support the statement I mentioned.

Author Response

Response to Comments from Reviewer #1:

Comment 1: Why is the study focused mostly on TFs, as other proteins like cell adhesion proteins, cytoskeletal proteins, signalling ligands, etc., also play crucial roles in cell differentiation?

Response: Thank you for the reviewer's concern. Indeed, besides transcription factors (TFs), other proteins, such as cell adhesion proteins, cytoskeletal proteins, signaling ligands, and more, also hold pivotal roles in cell differentiation. In our study, our primary focus was on TFs. This emphasis stemmed from the fact that Bacillus anthracis lethal toxin, a well-known metalloprotease, cleaves the N-terminal domain of all mitogen-activated protein kinase (MAPK) kinases (MKKs/MEKs) from MEK1 to MEK7, except for MEK5 [1]. The MAPK signaling pathway has demonstrated its role in regulating transcription in response to extracellular signals [2]. Hence, our attention primarily centered on the TFs impacted by the disruption of the MAPK signaling pathway in the microarray data we analyzed. For a clearer presentation, we have revised the sentence “Among these 365 genes, 14 encode transcription factors associated with cell differentiation and development.” to “From the pool of 365 genes, our primary focus was identifying transcription factors affected by the disruption of the MAPK signaling pathway caused by LT. Among these genes, we identified 14 that encode transcription factors associated with cell differentiation and development.” (Lines 157-160).

Comment 2: How did the authors determine that DACH1 is the critical gene among those 14 TFs that they found from the study? Please add relevant methodology.

Response: At the outset, we lacked information on the pivotal gene and conducted a series of shRNA knockdown assays targeting the 14 identified TFs individually. The purpose was to ascertain whether reducing the expression of these genes would suppress megakaryocytic differentiation in HEL cells. Subsequently, our investigations revealed the crucial role of the DACH1 gene in megakaryopoiesis, particularly in polyploidization. Notably, ZFP36L1 and FOSB, listed in Figure 5, were initially among the 14 TFs studied and are now recognized as known TFs. For a more precise presentation, we have revised the sentence “DACH1, identified as a critical gene in the cDNA microarray assay, is known for its role in eye and genital development in various species, including Drosophila, mice, and humans.” to “DACH1, one of the 14 transcription factors identified in the cDNA microarray assay, is recognized for its involvement in eye and genital development across various species, including Drosophila, mice, and humans.” (lines 160-162).

Comment 3: Within lines 395-396, overnight fixation of HEL cells at -20 °C in 70% ethanol may induce cell aggregation or affect expression. Do the authors have any documented protocol that they have followed?

Response: We followed the protocols established in previous studies [3-5] and identified the unaggregated portion when analyzing our data. Despite observing minimal cell aggregation per the established protocols, we specifically examined the unaggregated piece, even when some cell aggregation was present.

Comment 4: Within lines 433-434, it is stated about a MOI of 8 for virus spread. The MOI that is picked can affect how well transduction works and how much knockdown is done. It might be helpful to include details about how this MOI was found and whether pilot studies were conducted to improve transduction efficiency.

And, within lines 435-436, adding 10-8 M TPA on the seventh day is mentioned to induce megakaryocytic differentiation. It would be helpful to provide a brief rationale for the choice of TPA concentration and duration of treatment.

Response: In our pilot studies, we tested MOIs of 3, 8, and 30, finding that MOIs of 8 and 30 yielded the highest levels of transduction and knockdown efficiency; however, due to economic constraints associated with the lentiviruses carrying pLKO.1-shDACH1 purchased from the National RNAi Core Facility, we decided to proceed with an MOI of 8 for our study.

Our pilot studies infected cells with lentiviruses carrying the enhanced green fluorescence protein (EGFP) and the puromycin-resistant gene. Following the infection with an MOI of 8, we introduced 2 μg/ml of puromycin on the third day to select the infected cells. To ensure the elimination of un-fluorescent cells, we maintained the presence of puromycin for four days, resulting in approximately 98% of cells being GFP-positive. Consequently, on the seventh day, we treated the cells with TPA to initiate megakaryocytic differentiation. Based on insights from prior studies [6-8] and our pilot studies, we chose to induce megakaryocytic differentiation using a 10-8 M TPA treatment administered for three days to optimize results.

To make more clearer presentation, we modified the sentences “On the third day, HEL cells were treated with 2 μg/ml puromycin (InvivoGen) to select infected cells. After seven days, 10-8 M TPA was added to induce megakaryocytic differentiation in HEL cells.” to “After three days, HEL cells were treated with 2 μg/ml puromycin (InvivoGen) for four days to eliminate uninfected cells. Following this, 10-8 M TPA was added on the seventh day to induce megakaryocytic differentiation in HEL cells.” (lines 460-463).

Comment 5: With respect to Figure 5 and results section 2.5, the authors mentioned in lines 366-367 that “DACH1 gene may positively regulate the FOSB, ZFP36L1, RUNX1, FLI1, AHR, and GFI1B genes.”

  1. I) Why only those genes (with respect to Fig 5)?
  2. II) Explain more about how your results support the statement I mentioned.

Response:

  1. I) After identifying DACH1 as a novel TF associated with megakaryopoiesis, we aim to explore its relationship with other TFs uncovered through our microarray assay. Given the novelty of several TFs from our findings, we plan to specifically demonstrate the association between DACH1 and two known TFs identified within our microarray data. Simultaneously, we seek to investigate the correlation of DACH1 with established TFs known to play roles in megakaryopoiesis. We've selected specific TFs documented in review papers and aim to assess whether DACH1 is an early-determining gene in the process of megakaryopoiesis. To make more clearer presentation, we revised the sentence “As the DACH1 gene is a novel transcription factor in megakaryopoiesis, we aim to explore its relationship with other genes involved.” to “As the DACH1 gene represents a newly discovered transcription factor in megakaryopoiesis, our objective is to investigate its associations with other transcription factors (FOSB and ZFP36L1) identified through our microarray assay and established transcription factors (RUNX1, FLI1, AHR, GATA1, NFE2, and GFI1B) known to participate in megakaryopoiesis. This exploration aims to ascertain whether DACH1 plays a pivotal role as an early-determining gene in megakaryopoiesis” (lines 256-261).
  2. II) As mentioned in lines 261-266, our hypothesis suggests that genes positively regulated by DACH1 will exhibit downregulation upon DACH1 gene knockdown, while genes negatively regulated by DACH1 will show upregulation following DACH1 gene knockdown. Additionally, we anticipate that genes upstream or unrelated to the DACH1 gene will show no change in the expression level following the DACH1 gene knockdown. To assess this, we analyzed the relative fold change in mRNA expression levels of all genes within the shDACH1-TPA treated group, using the mRNA expression level of each gene in the shLacZ-TPA treated group as the reference (set at one-fold). The updated Figure 5 presents a clearer representation illustrating the relative expression levels of FOSB, ZFP36L1, RUNX1, FLI1, AHR, and GFI1B, which were observed to be downregulated upon DACH1 gene knockdown. This observation concludes that that DACH1 gene may positively regulate the FOSB, ZFP36L1, RUNX1, FLI1, AHR, and GFI1B To make more clearer presentation, we revised the sentence “Subsequent qRT-PCR analysis, our results suggested that the DACH1 gene may positively regulate the FOSB, ZFP36L1, RUNX1, FLI1, AHR, and GFI1B genes. Conversely, the GATA1 and NFE2 genes may be either upstream or unrelated to the DACH1 gene.” to “We conducted qRT-PCR analysis to assess the relative fold change in mRNA expression levels of all genes within the shDACH1-TPA treated group. We used the mRNA expression level of each gene in the shLacZ-TPA treated group as the reference (set at one-fold). Our findings indicate that the relative expression levels of FOSB, ZFP36L1, RUNX1, FLI1, AHR, and GFI1B were downregulated upon DACH1 gene knockdown. However, the relative mRNA expression levels of GATA1 and NFE2 remained unchanged following the DACH1 gene knockdown.” (lines 267-273).

References:

  1. Bardwell, A. J.; Abdollahi, M.; Bardwell, L., Anthrax lethal factor-cleavage products of MAPK (mitogen-activated protein kinase) kinases exhibit reduced binding to their cognate MAPKs. Biochem J 2004, 378, (Pt 2), 569-77.
  2. Whitmarsh, A. J., Regulation of gene transcription by mitogen-activated protein kinase signaling pathways. Biochim Biophys Acta 2007, 1773, (8), 1285-98.
  3. Yan, X. B.; Yang, D. S.; Gao, X.; Feng, J.; Shi, Z. L.; Ye, Z., Caspase-8 dependent osteosarcoma cell apoptosis induced by proteasome inhibitor MG132. Cell Biol Int 2007, 31, (10), 1136-43.
  4. Liu, R.; Pu, D.; Liu, Y.; Cheng, Y.; Yin, L.; Li, T.; Zhao, L., Induction of SiHa cells apoptosis by nanometer realgar suspension and its mechanism. J Huazhong Univ Sci Technolog Med Sci 2008, 28, (3), 317-21.
  5. Liang, M.; Hu, K., Involvement of lncRNA-HOTTIP in the Repair of Ultraviolet Light-Induced DNA Damage in Spermatogenic Cells. Mol Cells 2019, 42, (11), 794-803.
  6. Papayannopoulou, T.; Nakamoto, B.; Yokochi, T.; Chait, A.; Kannagi, R., Human erythroleukemia cell line (HEL) undergoes a drastic macrophage-like shift with TPA. Blood 1983, 62, (4), 832-45.
  7. Long, M. W.; Heffner, C. H.; Williams, J. L.; Peters, C.; Prochownik, E. V., Regulation of megakaryocyte phenotype in human erythroleukemia cells. J Clin Invest 1990, 85, (4), 1072-84.
  8. Ogura, M.; Morishima, Y.; Okumura, M.; Hotta, T.; Takamoto, S.; Ohno, R.; Hirabayashi, N.; Nagura, H.; Saito, H., Functional and morphological differentiation induction of a human megakaryoblastic leukemia cell line (MEG-01s) by phorbol diesters. Blood 1988, 72, (1), 49-60.

Reviewer 2 Report

Comments and Suggestions for Authors

In the present manuscript the authors have made attempts to communicate the outcome of their study in identifying DACH1 as a missing link in understanding megakaryocytic sup- 2

pression by Bacillus anthracis lethal toxin. They have utilized a commercially available a human erythroleukemia(HEL) cell line to perform  most of the studies reported here and also have presented data on  DACH1 gene expression in mononuclear cells isolated from healthy donors andpatients with thrombocytopenia. They have utilized (12-O-tetradecanoylphorbol-13-acetate) (TPA)-induced megakaryocytic differentiation in human erythroleukemia (HEL) cells to identify genes involved in LT-induced megakaryocytic suppression. They aimed to investigate the impact of lethal toxin (LT) on TPA-induced megakaryocytic differentiation in human erythroleukemia(HEL) cells. Overall the working hypothesis and the experimental strategies taken to support the idea to establish a working model for the study appears promising, but due to lack of clear understanding of the necessary experimental designs, techniques used, lack of appropriate controls, use of discontinued or outdated raw materials and wrongly performed statistical data analysis make the current report not compelling.

Comments to the editor: 

1. I recommend showing statistical data analysis as Results±SEMSupplementary fig S1: Why is at 100% the SD error bar visible. The untreated control value for the cell viability has been strategically forced to take a value of 100. The SEM for this data should be 0 with no apparent SD value.

2. Fig 2B: Where is TPA+LT data for WB? Show the normalized band intensity data as bar graph (Normalized DACH1 band intensity with respect to beta actin±SEM). How were the band signal intensities captured? How many times the experiments were repeated (missing n value)?

3. Figs 4A and B: No MEK-1 and 2 bands have been captured in the same membrane and hence not a valid way of presenting data. Same comments also hold for ERK1/2 blots. They have not been captured in the same membrane.Missing experimental repeats and data analysis. Beta actin, MEK5 and ERK2 band intensities appear supersaturated and does not seem to be suitable for running a true quantitative data analysis. Why was the tERK band intensity significantly higher (Fig 4B, 72h, Vehicle data)?

4. Fig 4C: With such a overlapping SD data for TPA and (TPA+U0126) group there should not exist any statistical significance between these two groups. Please redo all statistical data analysis employed in the present manuscript.

5. Fig 5: Where is the mRNA expression data for control or scrambled shRNA product? Without showing appropriate control data fig 5 does not make any sense to me. Overall this fig is very poorly presented, analyzed. Based on the present data, it should not be conclusively determined which genes upstream, downstream or unrelated to DACH1 and how are they linked to the megakaryocyticdifferentiation pathway in the HEL cell line.

6. Figure 7A: This table has been cited in the text as Fig 7A. Actually this needs to be cited separately as Table 1. The implication and significance of this table need to be elaborated in detailed as this table presents real patient and donor data.  

7. The WB methods section is pretty premature and full of lack of information. The authors need to show data providing details of individual steps implemented in the study. There has been both anti-mouse (ERK1/2, MEK1, beta actin), and anti-rabbit (pERK1/2, DACH1, MEK2) primary antibodies used but no detailed information of what secondary antibodies used in the study was available in the methods. How the band intensities were digitally captured and analyzed? DACH1 Rabbit polyclonal antibody from Abcam is a discontinued product because of the QC compliance issueAny claim of using this antibody in a study needs to be properly documented providing further evidence that the authors indeed used this antibody when the product was still compliant with QC related issue between production batches. Similarly MEK2 antibody from Santa Cruz Biotechnology is a discontinued product. Using non-compliant or discontinued product do not provide an opportunity to the readers to repeat the study as mentioned in the manuscript or does not provide any future possibility to plan on further experiments in the similar or allied studies. The supplementary data on WB study results are vague, obscure or poorly captured digital images with mol wt marker bands sitting separately on the membranes and do not help establishing the true band identity and hence not adding any values. 

 This 

Author Response

Response to Comments from Reviewer #2:

Comment 1: I recommend showing statistical data analysis as Results±SEM. Supplementary fig S1: Why is at 100% the SD error bar visible. The untreated control value for the cell viability has been strategically forced to take a value of 100. The SEM for this data should be 0 with no apparent SD value.

Response:
Thank you for the notification. We have updated our statistical data analysis to present mean values accompanied by the standard mean error (SEM). Furthermore, we have introduced several new figures (Fig-1B, Fig-1D, Fig-2A, Fig-2B, Fig-3A, Fig-3C, Fig-3E, Fig-4C, Fig-4D, Fig-4E, Fig-5, Fig-6, Fig-7, and Fig-S1) along with corresponding statistical representations. In Supplementary Fig S1, we have made the necessary modifications per the reviewer’s suggestions.

Comment 2: Fig 2B: Where is TPA+LT data for WB? Show the normalized band intensity data as bar graph (Normalized DACH1 band intensity with respect to beta actin±SEM). How were the band signal intensities captured? How many times the experiments were repeated (missing n value)?

Response: Thank you for the reviewer's reminder. We have incorporated the new Figure 2B, which displays the Western blot (WB) results for TPA+LT data. In addition, the normalized intensity data are presented as a bar graph with the times the experiments repeated. We also modified the sentence “Western blot analysis also showed an increase in DACH1 protein levels upon TPA treatment.” to “Western blot analysis also showed an increase in DACH1 protein levels upon TPA treatment and a decrease in DACH1 protein levels upon the combination of TPA with LT treatment.” (lines 167-169) and sentence “This data strongly supports the significant upregulation of both DACH1 mRNA and protein during megakaryocytic differentiation.” to “This data strongly supports the significant upregulation of both DACH1 mRNA and protein during megakaryocytic differentiation and the downregulation of both DACH1 mRNA and protein induced by LT treatment in the megakaryocytic process.” (lines 170-173).

For more clearer presentation, the sentence “The gel images were inverted using ImageJ software (1.53e) and cropped using Adobe Photoshop CC 2019.” to “The gel images were inverted using ImageJ software (version 1.53e), and then the band signal intensities were captured and quantified using ImageJ software. Subsequently, the gel images were cropped utilizing Adobe Photoshop CC 2019.” (lines 495-497).

Comment 3: Figs 4A and B: No MEK-1 and 2 bands have been captured in the same membrane and hence not a valid way of presenting data. Same comments also hold for ERK1/2 blots. They have not been captured in the same membrane. Missing experimental repeats and data analysis. Beta actin, MEK5 and ERK2 band intensities appear supersaturated and does not seem to be suitable for running a true quantitative data analysis. Why was the tERK band intensity significantly higher (Fig 4B, 72h, Vehicle data)?

Response: Thank you for your inquiry. Ideally, capturing the bands of the proteins of interest and the internal control proteins on the same membrane is preferable. However, it's challenging to capture all these bands on a single membrane due to the close molecular weights of MEK-1, MEK-2, β-actin, or p-ERK, and ERK. The repeated stripping steps required in such cases could potentially distort the results. Although we analyzed MEK-1, MEK-2, and β-actin, or p-ERK and ERK on separate membranes, we ensured equal protein amounts (30 μg) loaded into each well.

We have replaced Fig-4A and Fig-4B with new data from the 10-minute time point. This updated data is presented using a bar graph, indicating repetitions across multiple experiments. To observe the partially retained band of MEK-2 in the presence of TPA and LT, we increased the exposure time to approximately 2 to 3 minutes. Consequently, the bands of MEK-2 may appear oversaturated. The significantly higher intensity of the ERK band in the vehicle control remains unexplained, but this anomaly does not affect the results we have presented.

Comment 4: Fig 4C: With such a overlapping SD data for TPA and (TPA+U0126) group there should not exist any statistical significance between these two groups. Please redo all statistical data analysis employed in the present manuscript.

Response:
Thank you for the notification. We have finished all statistical data analyses as per the Results±SEM format specified in the manuscript. Additionally, we have replaced Fig-4C with the newly generated Fig-4E.

Comment 5: Fig 5: Where is the mRNA expression data for control or scrambled shRNA product? Without showing appropriate control data fig 5 does not make any sense to me. Overall this fig is very poorly presented, analyzed. Based on the present data, it should not be conclusively determined which genes upstream, downstream or unrelated to DACH1 and how are they linked to the megakaryocytic differentiation pathway in the HEL cell line.

Response:
Thank you for the notification. Our control group was established using shLacZ-TPA. Our analysis examined the relative fold change in mRNA expression levels of all genes within the shDACH1-TPA treated group, with each gene's mRNA expression level in the shLacZ-TPA treated group set as the reference (normalized to one-fold). We have updated the presentation style and introduced a new Figure 5, where each gene's relative fold mRNA expression after the DACH1 gene knockdown is depicted alongside a comparable bar representing the reference mRNA expression of the same gene in the shLacA-TPA control group.

Comment 6: Figure 7A: This table has been cited in the text as Fig 7A. Actually this needs to be cited separately as Table 1. The implication and significance of this table need to be elaborated in detailed as this table presents real patient and donor data.  

Response:
Thank you for your reminder. We have created Table 1 using the data previously presented in the old Fig. 7. Additionally, we have adjusted the citations accordingly in the revised manuscript, distinguishing Table 1 and Fig. 7 as separate entities. For the implication and significance of the table, we have added the following description. “Patients 1 and 2 (male) exhibit erythroid defects characterized by lower red blood cell (RBC) counts (normal range for men: 4.50–5.90 × 106/μL; normal range for women: 4.00–5.20 × 106/μL), reduced hemoglobin (Hb) levels (normal range for men: 13.5–17.5 g/dL; normal range for women: 12.0–16.0 g/dL), and decreased hematocrit (Hct) values (normal range for men: 41.0–53.0%; normal range for women: 36.0–46.0%), compared to three healthy donors [Donor 1 (female), Donors 2 and 3 (male)]. Additionally, all patients exhibit thrombocytopenia, characterized by lower platelet counts than healthy individuals (normal range for platelet count: 150–400 × 103/μL).” (lines 313-320).

Comment 7: The WB methods section is pretty premature and full of lack of information. The authors need to show data providing details of individual steps implemented in the study. There has been both anti-mouse (ERK1/2, MEK1, beta actin), and anti-rabbit (pERK1/2, DACH1, MEK2) primary antibodies used but no detailed information of what secondary antibodies used in the study was available in the methods. How the band intensities were digitally captured and analyzed? DACH1 Rabbit polyclonal antibody from Abcam is a discontinued product because of the QC compliance issue. Any claim of using this antibody in a study needs to be properly documented providing further evidence that the authors indeed used this antibody when the product was still compliant with QC related issue between production batches. Similarly MEK2 antibody from Santa Cruz Biotechnology is a discontinued product. Using non-compliant or discontinued product do not provide an opportunity to the readers to repeat the study as mentioned in the manuscript or does not provide any future possibility to plan on further experiments in the similar or allied studies. The supplementary data on WB study results are vague, obscure or poorly captured digital images with mol wt marker bands sitting separately on the membranes and do not help establishing the true band identity and hence not adding any values. 

Response:
Many thanks for the notification. We have modified the sentence “Subsequently, the membrane was incubated with a horseradish peroxidase-conjugated secondary antibody for 1 hour at room temperature“ in the Materials and Method section as “Afterward, the membrane probed with DACH1, MEK-2, and p-ERK, and the membrane analyzed with MEK-1, ERK, and β-actin were separately incubated with horseradish peroxidase-conjugated anti-mouse secondary antibody (dilution 1:10,000, #31430, Pierce) and anti-rabbit secondary antibody (dilution 1:5,000, #sc-2004, Santa Cruz) for 1 hour at room temperature.” (lines 486-491).

As response in Comment 2, we have added the sentences “The gel images were inverted using ImageJ software (version 1.53e), and then the band signal intensities were captured and quantified using ImageJ software. Subsequently, the gel images were cropped utilizing Adobe Photoshop CC 2019.” in the Materials and Methods section (lines 495-497).

Specifically, we procured the DACH1 antibody in March 2009 (lot number: #513999) and the MEK-2 antibody in May 2007 (lot number: #C1203). We believe that there are no quality control issues with these products at the time of purchase. Unfortunately, these antibodies are discontinued products, and readers cannot replicate the study using the same antibodies mentioned in the manuscript. However, I want to clarify that we utilized an alternative DACH1 antibody (ab176718) obtained from Abcam, which yielded results similar to those described in the manuscript.

The WB experiments were conducted several years ago. In Supplementary Fig-S2A, Fig-S2B, and Fig-S4A, we provided the membrane blots marked with pencil to indicate protein molecular weight markers. This annotation was necessary as we used unstained protein molecular weight markers. Unfortunately, the original membrane blots were not retrievable for Supplementary Fig-S3A, Fig-S3B, and Fig-S4B. In these instances, we annotated the relative molecular weight positions based on a reference pattern from another blot. This reference blot had known molecular weights marked and was probed with the same antibody.

Round 2

Reviewer 1 Report

Comments and Suggestions for Authors

The revised manuscript may be accepted for publication.

Author Response

Response to Comments from Reviewer #1:
Comment 1: The revised manuscript may be accepted for publication.

Response: Thank you for the valuable suggestions provided by the reviewer. These suggestions improve the readability and clarity of our manuscript.

Reviewer 2 Report

Comments and Suggestions for Authors

Thanks for making attempts for providing responses to the comments that I made on the manuscript based on my initial review. Unfortunately the responses provided here do not satisfactorily address my concerns.

1. Figure S1, untreated goup still shows visible error bars though the values have been strategically placed to 100% for each repeat and that alone should push the SD to 0 and eventually the resultant SEM  becomes 0 (SEM=SD/(n)^1/2).

2. All the WB related queries do not make me convinced on experimental design  total and phospho-band intensity data captured on different membranes), use of old antibodies (procured in 2007 and 2009 ), though the authors have mentioned running those blots >10 years ago which I do not think the case.

3. Overall the way the responses have been drafted indicated there is an eager attempt to comply with the task without providing any reasonable scientific and logical explanations.

Author Response

Response to Comments from Reviewer #2:

Comment 1: Figure S1, untreated group still shows visible error bars though the values have been strategically placed to 100% for each repeat and that alone should push the SD to 0 and eventually the resultant SEM becomes 0 (SEM=SD/(n)^1/2).

Response: Thank you for the reviewer’s suggestion. We have modified Figure S1 based on the reviewer’s request. We have changed the line point of the bar and SEM to make the figure more straightforward.

Comment 2: All the WB related queries do not make me convinced on experimental design total and phospho-band intensity data captured on different membranes), use of old antibodies (procured in 2007 and 2009 ), though the authors have mentioned running those blots >10 years ago which I do not think the case.

Response: We acknowledge the reviewer's comment. Ideally, the band intensities of p-ERK, ERK, or MEK1, MEK2, and β-actin should have been captured on the same membranes to accurately represent the fundamental signaling changes. Our Western blot experiments were conducted a dozen years ago. Unfortunately, we are uncertain if all these band intensities originated from the same blot, as we couldn't locate the blot images. Considering the time constraints for the revision, repeating all the Western blot experiments isn't feasible. Consequently, we are addressing the reviewer's suggestions based on the available data.

Comment 3: Overall the way the responses have been drafted indicated there is an eager attempt to comply with the task without providing any reasonable scientific and logical explanations.

Response: Regrettably, with the existing data available to us, we are unable to adequately address the reviewer’s suggestions regarding the Western blot. The scientific and logical explanations necessitate repeating the Western blot to display all relevant band intensities and molecular weight markers on a single membrane. We are willing to undertake this process; however, it requires additional time. We need to request lethal toxin, test the appropriate dilution fold, and acquire new antibodies to determine the optimal dilution fold for conducting the Western blot anew.

Round 3

Reviewer 2 Report

Comments and Suggestions for Authors

Thanks for making attempts for providing responses to the comments that I made on the manuscript based on my second review. Unfortunately the responses provided here related to the WB questions are still not satisfactorily addressed. Authors being unable to provide with any reasonable explanations for the source and details of raw data is a major drawback.